# PACE: Part-Wise Slow-Fast Conditioning for Dance-to-Music Generation

## Abstract

Dance-to-Music generation aims to compose music that is rhythmically aligned with human dance movements. While recent diffusion-based approaches have achieved promising results, they treat the dancer's body as a holistic unit when extracting motion features, thereby overlooking the fine-grained rhythmic contributions of individual body parts and the heterogeneous temporal dynamics manifested in both slow and fast motion patterns. In this work, we approach the dance-to-music generation task from a fresh conditioning encoding viewpoint, where part-wise motion energy decomposition and a hierarchical slow-fast conditioning encoder are integrated to generate the conditioning for music latent diffusion. Through comprehensive subjective and objective evaluations of rhythm synchronization and generated music quality, experimental results on the AIST++ and TikTok benchmarks confirm that our framework consistently outperforms existing state-of-the-art approaches for dance-to-music generation.

## 1 Introduction

Recent years have witnessed the unprecedented growth of user-generated content on short-video platforms (*e.g.*, TikTok[1] and YouTube[2]), facilitated by the ubiquity of mobile devices and the increasing demand for creative self-expression. Among various forms of content, human-centric dance videos have become one of the most prominent and widely shared media types, attracting substantial user engagement. As put by the renowned choreographer George Balanchine, "Dance is music made visible", a statement that underscores the intrinsic and inseparable relationship between music and bodily movement. In this context, the choice of background music plays a pivotal role in shaping the expressive quality of dance videos and substantially influencing the overall user experience. Accordingly, the intersection of computer vision and audio synthesis has given rise to the emerging task of dance-to-music (D2M) generation in both academic and industry areas (Gan et al., 2020; Di et al., 2021; Zhu et al., 2022a;b; Yu et al., 2023; Han et al., 2024; Li et al., 2024; Zhang & Hua, 2024; Liang et al., 2024; Sun et al., 2025; Ji et al., 2025), which aims to exploit deep generative AI techniques to automatically produce music that exhibits rhythmic and stylistic coherence with the given human-centric dance sequence.

According to the generative paradigms, existing D2M generation methods can be roughly classified into two major lines: autoregressive(Di et al., 2021; Han et al., 2024) and non-autoregressive modeling (Zhu et al., 2022b; Sun et al., 2025), where autoregressive approaches typically generate musical sequences frame by frame or token by token conditioned on previously generated outputs, while non-autoregressive ones generate musical segments in parallel, thereby avoiding sequential dependency. Among non-autoregressive approaches, diffusion-based D2M generative models have recently attracted increasing attention due to their strong capacity for modeling complex distributions and producing high-quality, diverse outputs. For example, Zhu et al. (2022b) improves the input-output correspondence and achieves higher or competitive general synthesis music quality by introducing a conditional discrete contrastive diffusion loss, conditioned on the motion and visual features extracted from human movement sequences and dance video frames through the corresponding motion and visual encoders. Meanwhile, Yu et al. (2023) designs a series of context-aware conditioning encoders to transform video frames, human poses, and categorical labels into visual embeddings, visual rhythm,

---

[1] https://www.tiktok.com/.
[2] https://www.youtube.com/.

and genre embeddings, and then hierarchically attend these conditionings into the audio diffusion model. Besides, Sun et al. (2025) adopts both positive rhythmic information and negative ones as conditionings to enhance the quality of generated music and its synchronization with dance videos in a dual-path diffusion structure. However, although these diffusion-based approaches have demonstrated notable progress in D2M generation, they share a common limitation regarding the dancer movements by modeling the dancer's body as a holistic unit, overlooking the fine-grained, part-level dynamics inherent in dancer movement. In fact, different body parts often follow distinct rhythmic patterns, where the torso may align with the global beat through smoother and more sustained dynamics, while the limbs frequently capture faster or stylistically distinct music variations. By investigating these patterns at a finer granularity, it becomes possible to uncover subtle but meaningful rhythmic signals that would otherwise be overshadowed by the stronger and dominant rhythmic signals from other body parts, thereby enabling a more comprehensive understanding of the dancer's overall rhythmic expression. Moreover, even within the same body part, individual joints often manifest heterogeneous temporal dynamics, where certain joints follow slower and smoother trajectories, whereas others display faster and more fine-grained motion patterns. Therefore, we ask:

> **(Q)** *Can decomposing motion into part-wise slow and fast components, and integrating them with joint-level semantic features, enable better alignment between heterogeneous dance movements and musical rhythms?*

To address **(Q)**, we study the diffusion-based D2M generative model from a fresh conditioning encoding viewpoint: **Part-wise slow-fAst Conditioning Encoding (PACE)**. Before diving the conditioning encoding process, we first present an illustrative example to analyze the essence of our proposed *part-wise slow-fast conditioning strategy*. Given a dance video, starting from 2D keypoints and reconstructed 3D poses, we partition the dancer body into five body parts (*e.g.*, Turso, Left Arm, Right Arm, Left Leg, and Right Leg), and obtain the "Slow" and "Fast" Kinetic energy components adopting a Butterworth filter(Shouran & Elgamli, 2020), respectively. As shown in **Fig. 1**, for the given dance video, the dancer's torso remains consistently active due to rotational movements and leg lifts. During the first 2.5 seconds, the predominant movement is localized to the right arm and right leg, indicating asymmetric engagement of the dancer's body. After

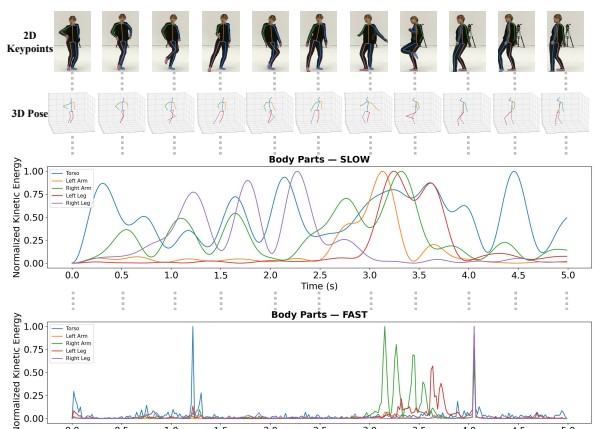

Figure 1: Illustration of our proposed part-wise slow-fast energy decomposition. Given a dance video, 2D keypoints are first obtained using AlphaPose (Fang et al., 2022) and 3D poses are then reconstructed through MotionBert (Zhu et al., 2023). The dancer body is partitioned into five body parts (torso, left/right arms, and left/right legs) and their corresponding normalized kinetic energy are computed in two frequency bands by applying a Butterworth filter (Shouran & Elgamli, 2020) to decompose the motion signals into slow and fast components.

approximately 2.5 seconds, the left arm and left leg are gradually activated, while the right arm exhibits noticeable swinging motions and the right leg primarily remains in a supporting stance with limited movement amplitude, resulting in a transition toward more coordinated and symmetric whole-body dynamics. Notable, after about 4 seconds, the entire body demonstrates a pronounced forward-backward swinging motion, highlighting a global shift in movement dynamics. Apparently, the variations of the slow and fast energy curves in **Fig. 1** are consistent with temporal evolution of the dance sequence. **On the one hand**, slow kinetic energy (Beeby et al., 2008) characterizes the low-frequency components of body motion, demonstrating the smooth, gradual, and sustained movements such as torso rotation, weight shifting, or steady limb positioning. It is evident that above temporal pattern of given dance video aligns with the upper plot of **Fig. 1**, which depicts the slow kinetic energy of each four body parts. During the initial 2.5 seconds, the trajectories corresponding to the left arm (orange) and left leg (red) remain flat, signifying minimal activity in these limbs. After 2.5 seconds, all curves exhibit pronounced fluctuations, reflecting the progressive activation of the left-side limbs in conjunction with the continuous torso movement and the swinging motions of the right arm. In contrast, the curve of the right leg tends to flatten after 2.5 seconds, indicating its role as

a supporting limb with limited dynamic variation. **On the other hand**, fast energy characterizes the high-frequency components of body motion, capturing transient and rapid movements such as limb swings, sudden lifts, and abrupt shifts in posture. As observed from the lower plot of the **Fig. 1**, after approximately 3 seconds, the right arm swing and the quick lift of the left leg give rise to prominent peaks in the corresponding curves. Meanwhile, a whole-body displacement occurs after 4 seconds, and curves of five body parts exhibit peak values simultaneously, reflecting a globally coordinated burst of motion. In the light of this, existing studies that treat the dancer's body as a holistic unit overlook such fine-grained temporal and spatial heterogeneity across different body parts.

To address the aforementioned limitations, we propose a novel diffusion-based dance-to-music generation framework that explicitly incorporates **Part-Wise Motion Energy Decomposition (PMED)** and **Hierarchical Slow-Fast Conditioning Encoder (HSF-Encoder)**, as illustrated in **Fig. 2**. Specifically, given a dance video, we first extract 2D keypoints and reconstruct 3D poses, which are then processed through PMED Module. Then, the resulting part-wise fast-slow temporal features are fed into HSF-Encoder. Finally, the obtained conditionings are fed into the **Music Latent Diffusion** module to synthesize music that is rhythmically aligned with the fine-grained dynamics of the input dance sequence. We summarize **our contributions** below.

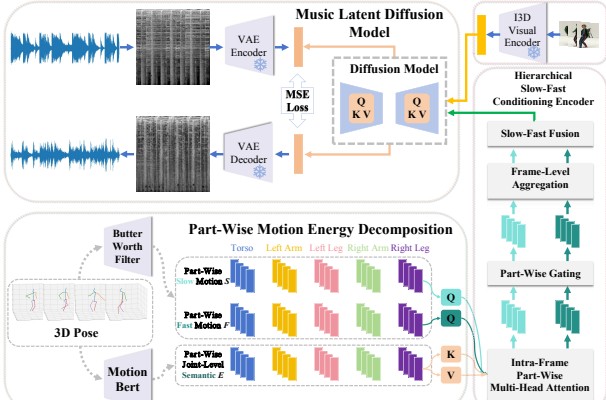

Figure 2: Overview of our diffusion-based D2M generation framework with PACE, which includes part-wise motion energy decomposition module and hierarchical slow-fast conditioning encoder to guide music latent diffusion model for rhythmically aligned musical audio generation.

- By decomposing motion energy into slow and fast components, we disentangles smooth and global temporal rhythm from transient and localized bursts of motion, enabling the diffusion-based D2M generation model to capture dance dynamics at multiple temporal scales.

- We provide fine-grained interpretability of dance temporal rhythm by revealing how different body parts contribute distinct temporal roles, which is rarely achievable when modeling the body as a holistic unit.

- We supply diffusion models with part-wise slow-fast motion features, providing complementary rhythmic cues to enhance the alignment of generated music with both macro-level (slow) and micro-level (fast) structures of dance.

- We conduct extensive experiments on two dance video datasets, providing both quantitative and qualitative analyses to demonstrate the competitiveness of our proposed PACE.

## 2 RELATED WORK

According to the generative paradigms, existing dance-to-music generation methods can be roughly divided into two categories: Autoregressive and Non-autoregressive modeling.

**Autoregressive Dance-to-Music Generation Modeling.** This branch (Huang & Yang, 2020; Aggarwal & Parikh, 2021; Su et al., 2021; Han et al., 2024; Liang et al., 2024) commonly predicts music tokens (*e.g.*, MIDI events and discrete audio token) sequentially, conditioned on previously generated outputs and features derived from the dance videos. For example, Aggarwal & Parikh (2021) propose a preliminary approach for dance-to-music generation, where music is represented as a sequence of notes from the C major pentatonic scale and dance is processed as a series of human poses extracted from video frames. To translate human body movements into rhythmic music, Su et al. (2021) take skeleton keypoints extracted from the dance videos as input and follow a sequence of models to generate synchronized soundtracks, where a transformer model is employed to generate drum hits and a U-Net model is introduced to predict the velocity and timing offsets of instruments. Besides, to realize the dance-driven multi-instrument music generation, Han et al. (2024) design a BERT-like (Koroteev, 2021) multi-instrument music generation model, where a Graph Convolutional

Network is adopted to encode dance motion and style from pose sequences and a Transformer with cross-attention is introduced to decode drum track sequences and capture rhythmic alignment. In a sense, although existing autoregressive D2M generation models excel at capturing long-term temporal dependencies and fine-grained structures, their sequential nature often incurs slow inference speed and limits scalability to long-form audio generation.

**Non-autoregressive Dance-to-Music Generation Modeling.** In contrast, non-autoregressive approaches (Zhu et al., 2022a;b; Yu et al., 2023; Tan et al., 2023; Zhang & Hua, 2024; Sun et al., 2025) work on generating music segments in parallel, thereby avoiding the step-by-step dependency of autoregressive decoding. For example, Zhu et al. (2022a) design an adversarial multi-modal framework and utilize the vector quantized audio representation, where video frames and human body motions are taken as conditioning input to guide the produced music plausibly aligns with the movements. Notably, diffusion models have emerged as powerful non-autoregressive paradigms for D2M generation, owing to their strong capability to model complex multi-modal distributions and corresponding ability to produce high-quality and diverse music samples. For example, Tan et al. (2023) establishe the first attempt to generate dance music directly from 3D human motion data with genre conditioning, where a UNet-based latent diffusion model is combined with pre-trained VAE to generate plausible dance music aligning with dynamic movements. Building on the success of diffusion for cross-modal synthesis, Sun et al. (2025) introduce PN-Diffusion, which enhances dance-music synchronization by incorporating both positive rhythmic information and negative rhythmic cues as conditionings in the designed dual diffusion and reverse processes. Overall, compared to autoregressive frameworks, diffusion-based non-autoregressive models provide higher flexibility and improve the ability to capture diverse rhythmic and stylistic variations.

In spite of the compelling success achieved by these diffusion-based methods in general cases, far too little attention has been paid to the fine-grained, part-level dynamics of human movement since most existing approaches treat the dancer's body as a holistic unit. In fact, different body parts often exhibit distinct rhythmic patterns and heterogeneous temporal dynamics, which are crucial for achieving precise alignment between dance movements and generated musical rhythms.

## 3 Preliminaries on Music Modality and Problem Formulation

**Mel-Spectrogram Music Representation.** For the musical audio part, while containing rich expressive and fine-grained temporal details, raw music waveforms are inherently high-dimensional and computationally expensive to process directly. To address this, musical audio is typically transformed into time-frequency representations such as Mel-spectrograms (Ustubioglu et al., 2023), which compress the signal while preserving perceptually relevant information. The Mel-spectrograms not only provide structured representation of temporal and spectral dynamics but also facilitate alignment with motion features that exhibit similar temporal patterns. By operating in this compact representation space, the complexity of the D2M generation problem is significantly reduced, enabling models to focus on meaningful correlations rather than redundant raw signal details. Suppose that we have a dance video $\mathcal{V} = \{\mathcal{M}, \mathcal{D}\}$, where $\mathcal{D}$ denotes the dance video consisting of $T$ frames. For the raw musical audios, we convert the input musical audio into Mel-spectrograms, with a sampling rate of 22,050 Hz and a Mel filter bank size of 256, resulting in a audio spectrogram $\mathcal{A}$, where the dimension of $A$ is $256 \times 256$. In this way, the complexity of the problem can be reduced while facilitating the learning of structured correspondences between dance movements and musical signals.

**Latent-Space Audio Modeling** Unlike conventional diffusion models (Ho et al., 2020; Nichol & Dhariwal, 2021) that rely on modeling high-dimensional raw data, Latent Diffusion Models (LDMs) (Rombach et al., 2022; Croitoru et al., 2023) conduct the generative process within a compressed latent space. As shown in the upper-left block of the **Fig. 2**, we introduce VAE encoder (Kingma et al., 2019) and VAE decoder to process the spectrograms into the latent space. In our real implementation, we utilize the music waveforms of the training dance videos to train the VAE encoder and decoder through the combination of a perceptual loss Zhang et al. (2018) and a patch-based adversarial objective Yu et al. (2021). Once trained, the VAE parameters are frozen, and the encoder-decoder pair is subsequently employed as a fixed component during the training of the diffusion model. Formally, given the audio spectrogram $A_i \in [0, 1]^{256 \times 256}$, the encoder $\mathcal{E}$ encodes $A_i$ into a latent representation $Z_i = \mathcal{E}(A_i)$, and the decoder $\mathcal{D}$ reconstructs the image from the spectrogram latent space $\tilde{A}_i = \mathcal{D}(Z_i)$, where $Z_i \in \mathbb{R}^{32 \times 32}$. This strategy ensures that the diffusion process operates

within a stable latent space while avoiding potential interference or instability that may arise from jointly optimizing both modules.

**Problem Formulation.** The objective of our D2M generation is to learn a conditional diffusion model as,

$$p_\theta(Z|S, F, E), \tag{1}$$

where a music spectrogram latent embedding $Z$ is optimized by using part-wise slow motion $S$, fast motion $F$, and part-wise joint-level semantic features $E$ as conditioning.

## 4 METHODOLOGY

### 4.1 PART-WISE MOTION ENERGY DECOMPOSITION

To enable rhythmically alignment, it is crucial to design motion representations that capture part-wise rhythmic nuances of human movement. To solve the limitation that conventional holistic modeling of the body often fails to account for the heterogeneous temporal dynamics exhibited by different body parts, we introduce the Part-wise Motion Energy Decomposition, including 3D pose extraction, part-wise slow-fast motion decomposition, and part-wise joint-level semantic encoding.

**3D Pose Extraction.** Intuitively, although 2D keypoints can represent skeletal trajectories in the image plane, they inherently lack depth information and are unable to fully characterize forward-backward displacements or subtle body rotations. Therefore, in our work, we first extract the frame-level 2D skeletons of dancer reagrding the dance video $\mathcal{D}$ using AlphaPose (Fang et al., 2022). Formally, given a dance video consisting of $N$ frames, we have $J^{2D} \in \mathbb{R}^{N \times 24 \times 3}$, where 24 corresponds to the number of body keypoints and $N$ refer to the dance video frame number, and each keypoint is represented by a 3-dimensional vector encoding its spatial position together with the associated confidence score. Then, inspired by the huge success of the MotionBert (Zhu et al., 2023) in 3D pose estimation, we reconstruct the 3D skeletal joints $J^{3D} \in \mathbb{R}^{N \times 17 \times 3}$, where 17 and 3 refer to the number of reconstructed 3D keyjoints and three spatial coordinates $(x, y, z)$ of each body keypoint, respectively.

**Part-Wise Slow-Fast Motion Energy Decomposition.** Typically, most existing approaches take the dancer's body as a holistic unit, overlooking the fact that the torso, arms, and legs often follow distinct temporal rhythms, as can be seen in **Fig. 1**. Generally, the torso tends to align with global beat patterns through smoother and sustained motions, while the arms and legs frequently capture finer rhythmic accents via rapid and localized movements. Beyond that, even within the same body part, individual joints can exhibit heterogeneous dynamics, where some follow slow and gradual trajectories while others display fast and fine-grained oscillations. Therefore, it is desirable to fully explore the motion diversity at both the part level and joint level, and hence realize the optimal alignment between dance movement and music rhythms.

Towards this end, we introduce the *Part-Wise Slow-Fast Motion Decomposition* strategy, which explicitly disentangles motion signals into frequency-specific energy components across different body parts. Firstly, regarding previously obtained 3D skeletal joints $J^{3D} \in \mathbb{R}^{N \times 17 \times 3}$, we partition dancer body into five body parts (*e.g.*, torso, left arm, left leg, right arm, and right leg), and obtain the part-wise 3D skeletal joints $J^{3D} = \{J^{3D}_{to} \in \mathbb{R}^{N \times 5 \times 3}, J^{3D}_{la} \in \mathbb{R}^{N \times 3 \times 3}, J^{3D}_{ll} \in \mathbb{R}^{N \times 3 \times 3}, J^{3D}_{ra} \in \mathbb{R}^{N \times 3 \times 3}, J^{3D}_{rl} \in \mathbb{R}^{N \times 3 \times 3}\}$. Then, we feed these 5 part-wise 3D skeletal joints to a Butterworth filter (Shouran & Elgamli, 2020) to decompose these body temporal movement signals into two frequency bands: (a) *slow band*, capturing low-frequency dynamics such as torso rotation and gradual limb positioning, and (b) *fast band*, characterizing high-frequency dynamics such as sharp arm gestures or rapid leg kicks. Finally, we have part-wise slow motion $S = \{S_{to}, S_{la}, S_{ll}, S_{ra}, S_{rl}\} \in \mathbb{R}^{N \times 1}$ and part-wise fast motion $F = \{F_{to}, F_{la}, F_{ll}, F_{ra}, F_{rl}\} \in \mathbb{R}^{N \times 1}$.

**Part-Wise Joint-Level Semantic Encoding.** While frequency-based motion energy decomposition captures rhythmic intensity, it lacks the ability to encode higher-level semantics of joint coordination that preserves rich contextual and temporal dependencies. Towards this end, we adopt the pre-trained human-centric motion encoder of MotionBert (Zhu et al., 2023) to obtain more powerful joint-level semantic representations for each body part. In particular, after feeding part-wise 3D skeletal joints to

the motion encoder, we have part-wise semantic features $E = \{E_{to}, E_{la}, E_{ll}, E_{ra}, E_{rl}\} \in \mathbb{R}^{N \times 512}$. In a sense, $E$ provide semantically enriched features that complement the part-wise slow-fast motion $S$ and $F$, enabling a more comprehensive conditioning representation for dance-to-music generation.

### 4.2 HIERARCHICAL SLOW-FAST CONDITIONING ENCODER

To transform part-wise slow and fast motion energies and joint-level semantic features into conditioning representations for diffusion-based music generation, as the major novelty, we introduce the Hierarchical Slow-Fast Conditioning Encoder, which consists of intra-frame part-wise multi-head attention, part-wise gating, frame-level aggregation, and slow-fast fusion components.

**Input Preparation.** *(a) Projection of Semantic Features.* While high-dimensional embeddings contain rich semantic information, the high-dimensional semantic features $E$ are not optimal for direct integration into downstream conditioning due to redundancy and the mismatch with the target feature space of D2M diffusion model. To this end, we first process the semantic features $E = \{E_{to}, E_{la}, E_{ll}, E_{ra}, E_{rl}\} \in \mathbb{R}^{T \times 512}$ as follows,

$$E_p' = f_{512 \to d}(\text{LN}(E_p)) \in \mathbb{R}^{N \times d}, \tag{2}$$

where $p \in \{to, la, ll, ra, rl\}$, $f_{512 \to d}$ refer to the linear projection layer, and LN denotes the Layer Normalization.

*(b) z-score Normalization.* To eliminate the magnitude bias and ensure consistent scaling across video sequences, we apply the z-score normalization to the slow-fast motion and feed them into MLP layers as follows,

$$S_p' = \text{MLP}(\text{zscore}(S_p)) \in \mathbb{R}^{N \times d}, F_p' = \text{MLP}(\text{zscore}(F_p)) \in \mathbb{R}^{N \times d}, \tag{3}$$

where $p \in \{to, la, ll, ra, rl\}$. In a sense, the normalized slow-fast motion highlights meaningful temporal dynamics, enabling the model to more effectively align heterogeneous dance movements with musical rhythms.

**Intra-Frame Part-Wise Multi-Head Attention.** As stated earlier, the obtained slow-fast motion captures the rhythmic intensity at different frequency bands, while the semantic features contain rich contextual and structural information about the dancer's pose (*e.g.*, coordination, articulation, and joint dependencies). To explicitly model the interactions among different body parts and ground rhythmic signals in meaningful joint-level semantics, we introduce the intra-frame part-wise multi-head attention, where slow and fast motions play the role of query, respectively, and semantic features are placed as Key and Value. Essentially, instead of treating all slow-fast motion features equally, we expect to utilize a network to dynamically learn how different body parts influence each other under slow and fast motion dynamics. In this way, slow motion can be used to highlight the stable body parts, and fast motion can be adopted to emphasize the rapid gestures. For $n$-frame, the part-wise multi-head attention is performed as follows,

$$
\begin{aligned}
H_{p,n}^s &= \text{MHA}_{p,n}(W_q^s(S_{p,n}'), W_k^s(E_{p,n}'), W_v^s(E_{p,n}')) \in \mathbb{R}^d, \\
H_{p,n}^f &= \text{MHA}_{p,n}(W_q^f(F_{p,n}'), W_k^f(E_{p,n}'), W_v^f(E_{p,n}')) \in \mathbb{R}^d,
\end{aligned}
\tag{4}
$$

where $W_q^s, W_k^s, W_v^s, W_q^f, W_k^f, W_v^f$ denote corresponding learnable projection matrices.

**Part-Wise Gating and Frame-Level Aggregation.** Intuitively, the torso may sometimes dominate through smooth global motions, whereas at other times the arms or legs may play a more significant role through rapid and accentuated movements. Therefore, different body parts contribute unequally to rhythmic expression at different time steps, and simply averaging or concatenating operations fail to reflect such dynamic importance. Towards this end, we introduce part-wise gating and frame-level aggregation to adaptively learns the relative importance of each anatomical part per frame and unify the information from multiple body parts into a single holistic representation for each frame. The details are as follows,

$$H_n^s = \sum_p g_{p,n}^s H_{p,n}^s \in \mathbb{R}^d, \quad H_n^f = \sum_p g_{p,n}^f H_{p,n}^f \in \mathbb{R}^d, \tag{5}$$

where $g_{p,n}^s$ and $g_{p,n}^f$ refer to the part-wise gate with regard to frame $n$.

**Slow-Fast Fusion.**    Above, we have obtained the slow and fast motion representations capturing complementary aspects of dance dynamics, where the *slow branch* reflects smooth and stable temporal patterns, and the *fast branch* emphasizes rapid and transient movements. However, if these two branches are processed independently, the conditioning signals will be fragmented since how these two rhythms interact within the same time frame is unknown. Thus, we obtain a unified conditioning representation by introducing a lightweight gating mechanism. Formally, given the aggregated representations $H_n^s$ and $H_n^f$, the fusion process is defined as,

$$\alpha_n = \sigma\big(f_{d \to d}(\text{LN}([H_n^s; H_n^f]))\big), \tag{6}$$

$$H_n = \alpha_n H_n^s + (1 - \alpha_n H_n^f), \tag{7}$$

where $\alpha_n$ denotes the learned gate that adaptively balances slow and fast contributions. Finally, by averaging over all inter-frame representations, we obtain the final diffusion conditioning $H$.

**Visual Conditioning**    Notably, beyond skeletal trajectories, we also incorporate I3D (Inflated 3D ConvNet Yadav & Kumar (2022))-based visual conditioning $G$ to complement motion features with richer temporal dynamics and visual context. In the real implementation, we concatenate them together as $C = [H, G]$.

### 4.3    MUSIC LATENT DIFFUSION

After obtaining the part-wise slow-fast conditioning $H$, we inject it into the U-Net backbone of the diffusion model. The details of the latent diffusion can be found in Appendix A. At each timestep $t$, the model predicts the noise as $\epsilon_\theta(Z_t, t, C)$. Meanwhile, the latent diffusion model can be trained via denoising objective function as,

$$\mathbf{L}_\epsilon = \mathbb{E}_{\epsilon \sim N(0,I), Z_t, t} \left\| \epsilon - \epsilon_\theta(Z_t, t, C) \right\|_2^2. \tag{8}$$

## 5    EXPERIMENT

### 5.1    DATASETS

We conduct experiments on two dance video benchmark datasets: AIST++ (Tsuchida et al., 2019) and TikTok (Zhu et al., 2022a). For a fair comparison with prior work, we strictly follow the dataset partitions and evaluation protocols adopted in Sun et al. (2025).

**AIST++.** It consists of 1,020 videos across 10 dance genres, each paired with its corresponding musical style. All recordings were captured in professional studio environments with clean backgrounds and calibrated camera poses, making it suitable for controlled evaluation. The music tracks cover 10 distinct categories (*e.g.*, lock, pop and breaking), with 6 pieces for each style. In our experiments, we use the official training/validation/testing split with 980, 20, and 20 videos, respectively. Following Sun et al. (2025), each video is segmented into 5-second clips, resulting in a total of 20,140 training instances and 234 testing instances.

**TikTok.** It is curated from real-world short video content, containing 445 dance clips paired with 85 unique songs. Following Sun et al. (2025), we divide the dataset into 392 videos for training and 53 for testing. In the same manner as AIST++ dataset, we also segment each video into 5-second clips, yielding 775 training and 103 testing samples. Compared with AIST++, TikTok provides more in-the-wild scenarios, making it a complementary resource for testing model generalization.

### 5.2    BASELINES AND EVALUATION PROTOCOLS

**Baselines.**    In our work, we include six recent state-of-the-art approaches: FoleyMusic (Gan et al., 2020), D2M-GAN (Zhu et al., 2022a), CMT (Di et al., 2021), CDCD (Zhu et al., 2022b), LORIS (Yu et al., 2023), and PN-Diffuison (Sun et al., 2025). These methods are selected because they are highly relevant to our D2M problem setting and provide publicly available implementations, pretrained weights, and hyper-parameters, which ensures reproducibility and fair comparison.

**Evaluation Protocols.**    (a)*Objective Evaluation.* To evaluate the alignment between generated music and dance movements, following prior works (Zhu et al., 2022b; Sun et al., 2025), two core metrics are the **Beats Coverage Score (BCS)** and **Beats Hit Score (BHS)**. Additionally, we also report **F1**

| Dataset | Method | BCS↑ | CSD↓ | BHS↑ | HSD↓ | F1↑ | FAD_v↓ | FAD_p↓ | FAD_c↓ |
|---------|--------|------|------|------|------|-----|--------|--------|--------|
| AIST++ | FoleyMusic (Gan et al., 2020) | 92.00 | 13.33 | 85.63 | 18.87 | 88.70 | 8.01 | 19.50 | 1.10 |
| | D2M-GAN (Zhu et al., 2022a) | 88.67 | 10.49 | 82.73 | 16.86 | 85.60 | 11.29 | 27.76 | 1.48 |
| | CMT (Di et al., 2021) | 95.92 | 8.19 | 61.70 | 24.66 | 75.41 | 12.57 | 13.24 | 1.02 |
| | CDCD (Zhu et al., 2022b) | 92.18 | 14.66 | 80.50 | 21.16 | 85.95 | 7.47 | 18.06 | 1.25 |
| | LORIS (Yu et al., 2023) | 95.84 | 7.89 | 95.09 | 16.09 | 96.45 | 7.71 | 50.27 | 0.77 |
| | PN-Diffusion(Sun et al., 2025) | 97.64 | 5.85 | 99.31 | 4.48 | 98.46 | 6.32 | 4.35 | 0.65 |
| | **PACE (Ours)** | 98.39 | 5.01 | 99.36 | 4.63 | 98.67 | 5.18 | 3.81 | 0.55 |
| TikTok | D2M-GAN (Zhu et al., 2022a) | 83.22 | 30.03 | 80.45 | 30.66 | 81.81 | 27.30 | 13.26 | 1.46 |
| | CMT (Di et al., 2021) | 85.42 | 32.56 | 60.03 | 31.07 | 70.52 | 20.45 | 15.56 | 1.30 |
| | CDCD (Zhu et al., 2022b) | 85.66 | 27.23 | 85.83 | 27.17 | 85.75 | 26.53 | 3.07 | 1.11 |
| | PN-Diffusion (Sun et al., 2025) | 89.51 | 17.11 | 91.73 | 13.33 | 90.60 | 16.37 | 1.14 | 1.25 |
| | **PACE (Ours)** | 91.31 | 14.20 | 92.10 | 11.23 | 91.50 | 14.25 | 1.11 | 1.05 |

Table 1: The quantitative comparison between PACE and baseline music generation models conducted on both AIST++ and TikTok testing set.

**scores** as well as the standard deviations of BCS and BHS, referred to as **CSD** and **HSD** to assess stability. Beyond rhythm metrics, we also introduce the **Fréchet Audio Distance (FAD)** (Kilgour et al., 2018) to measure distribution-level similarity between generated and original paired music, where three feature extractors: VGGish (Diwakar & Gupta, 2024) (FAD_v), PANNs (Kong et al., 2020) (FAD_p), and CLAP (Elizalde et al., 2023) (FAD_c) are adopted simultaneously.

(b)*Subjective Evaluation.* Same as existing work (Sun et al., 2025), we conduct a user study on AIST++ samples, where volunteers are invited to rate each generated music sample from 1 to 5 for overall quality (**OVL**) and cross-modal relevance to the dance video (**REL**). The averaged scores yield the Mean Opinion Score (MOS) (Streijl et al., 2016) and higher MOS indicates better quality and alignment. We also perform a Turing Test, asking volunteers to distinguish between generated and real music. The proportion of generated samples identified as real reflects perceptual realism.

**Implementation Details.** All musical audios are sampled at 22,050 Hz and segmented into 5-second segments, and the Mel-spectrogram resolution is 256. During training,we use a batch size of 64 and train the model for 200 iterations with 1,000 diffusion steps for inference. The model contains 159.66M parameters, and training one epoch on AIST++ requires approximately 7 minutes on a single NVIDIA RTX A6000 GPU.

## 5.3 MODEL COMPARISON

To thoroughly assess the cross-modal correspondence between dance videos and generated music, in **Tab. 1**, we report quantitative results across eight evaluation metrics. We highlight two key observations below. **First**, our proposed PACE consistently outperforms other baselines on both AIST++ and TikTok datasets, highlighting the advantage of the PACE in modeling the temporal rhythm of the musical audio. By incorporating part-wise slow-fast motion conditioning, PACE enables a more fine-grained representation of body movements, thereby capturing joint-level temporal rhythms that are closely aligned with musical cues. **Second**, in terms of FAD_v, FAD_p and FAD_c, the performance of PACE is significantly better than all baselines and the numerical results are largely smaller than the baseline methods, which confirms the effectiveness of our designed part-wise slow-fast conditioning strategy. Besides, the user study can be found in Appendix B.

## 5.4 ABLATION STUDY

To better explain the benefit of incorporating I3D conditioning features, we conducted the comparative experiment with two derivative of our model, namely, "PACE-G" and "PACE-H", where only visual conditioning $G$ or hierarchical slow-fast conditioning is adopted as the supervision of the music latent diffusion model. Meanwhile, we also explored the scenarios where either the slow motion or the fast motion branch is individually combined with visual conditioning, denoted as "PACE-Slow-G" and "PACE-Fast-G", respectively. These comparative settings allow us to disentangle the contributions of different conditioning signals and demonstrate the advantage of our full PACE framework. Moreover, we explored a variant termed "PACE-concat", where the slow motion features and fast motion features are directly concatenated with joint-label semantic encoding as the conditioning input. The comparative experimental results for these five variants of PACE can be found in **Tab. 2**. As can be seen, **(i)** PACE consistently outperforms other model variants, which well validates the necessity of taking into account the part-wise slow-fast motion conditioning and I3D

| Dataset | Method | BCS↑ | CSD↓ | BHS↑ | HSD↓ | F1↑ | FAD_v↓ | FAD_p↓ | FAD_c↓ |
|---------|--------|------|------|------|------|-----|--------|--------|--------|
| AIST++ | PACE-G | 91.60 | 15.67 | 85.30 | 23.42 | 88.34 | 9.13 | 15.25 | 1.65 |
| | PACE-H | 98.13 | 5.29 | 99.23 | 5.01 | 98.66 | 6.19 | 5.50 | 2.35 |
| | PACE-concat | 95.84 | 7.20 | 96.02 | 7.14 | 95.93 | 7.56 | 8.10 | 2.81 |
| | PACE-Slow-G | 97.92 | 5.54 | 98.20 | 7.53 | 98.06 | 6.15 | 7.20 | 2.50 |
| | PACE-Fast-G | 96.72 | 6.63 | 98.54 | 6.65 | 97.41 | 6.14 | 7.14 | 2.10 |
| | PACE | 98.39 | 5.01 | 99.36 | 4.63 | 98.67 | 5.18 | 3.81 | 0.55 |
| TikTok | PACE-G | 84.10 | 28.75 | 81.05 | 29.80 | 82.40 | 25.80 | 12.90 | 1.42 |
| | PACE-H | 86.25 | 26.90 | 83.15 | 28.10 | 84.47 | 21.75 | 14.85 | 1.28 |
| | PACE-concat | 84.55 | 28.00 | 81.10 | 27.40 | 82.79 | 24.58 | 16.12 | 1.59 |
| | PACE-Slow-G | 85.95 | 25.80 | 84.55 | 26.40 | 85.10 | 24.60 | 3.25 | 1.14 |
| | PACE-Fast-G | 88.35 | 18.20 | 91.20 | 14.25 | 90.15 | 17.10 | 1.28 | 1.21 |
| | PACE | 91.31 | 14.20 | 92.10 | 11.23 | 91.50 | 14.25 | 1.11 | 1.05 |

Table 2: Ablation on the diffusion conditioning.

visual conditioning. **(ii)** Notably, we found that "PACE-H" also achieves competitive performance performance compared to the baselines in **Tab. 1**, demonstrating the effectiveness of hierarchical slow-fast motion conditioning even without additional visual cues. **(iii)** While this simple fusion strategy provides a straightforward way of combining different temporal dynamics, it fails to fully exploit the complementary relationships and hierarchical dependencies between slow and fast motion signals. In contrast, our proposed hierarchical slow-fast conditioning encoder adaptively balances and integrates these two branches at multiple levels, thereby capturing more fine-grained rhythmic correlations between dance movements and musical audio.

## 5.5 VISUALIZATION OF RHYTHM CURVES

While quantitative metrics such as BCS, BHS, and FAD provide an objective evaluation of music generation quality, they do not directly reveal how well the generated music captures the fine-grained rhythmic structures present in dance videos. To bridge this gap, we visualize the rhythmic curves of generated music by plotting the normalized RMS energy and onset strength over time, which reflects the amplitude envelope of the music audio signal, and characterizes the local temporal changes and beat positions, respectively. In a sense, such visualization can qualitatively examine whether the rhythmic fluctuations of generated music are consistent with the temporal dynamics of the original dance video paired music. As shown in **Fig. 3**, we select the competitive baseline PN-Diffuison as comparison. **i** For PN-Diffusion (middle row), although the generated curves capture certain rhythmic fluctuations, they often exhibit

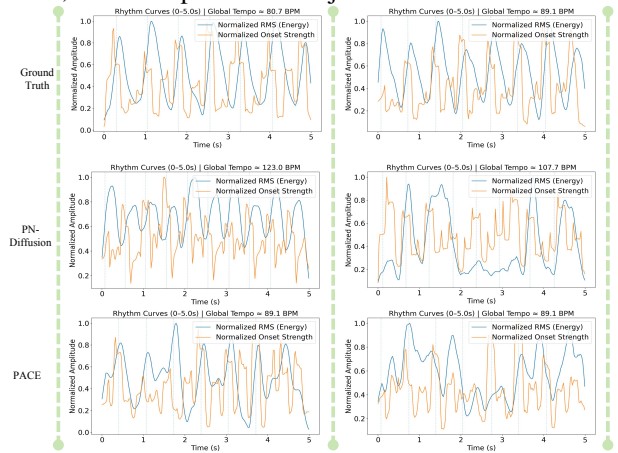

Figure 3: Visualization of rhythmic curves (normalized RMS energy and onset strength) for original paired music, PN-Diffusion, and our proposed PACE.

noisy patterns with excessive oscillations, leading to unstable alignment with the dance dynamics. In contrast, our PACE model (bottom row) produces rhythm curves that are smoother and more coherent, with onset peaks better synchronized with energy rises. **ii** the global tempo of music generated by PACE is closer to the ground truth compared to PN-Diffusion, highlighting the effectiveness of part-wise slow-fast conditioning in modeling fine-grained dance rhythms.

## 6 CONCLUSION

In this paper, we introduce PACE, a novel part-wise slow-fast conditioning encoder for dance-to-music generation. Unlike prior approaches that treat the dancer's body as a holistic unit, we explicitly decomposes part-wise motion signals into slow and fast dynamics. Meanwhile, we introduce a hierarchical slow-fast conditioning encoder to transform part-wise slow and fast motion energies and joint-level semantic features into conditioning representations for diffusion-based music generation. Our method demonstrates superior performance over two widely-used dance video datasets through objective and subjective evaluations. We refer readers to Appendix C for the details of LLM usage.

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

APPENDIX

## A  LATENT DIFFUSION MODEL

In this work, we aim to devise an end-to-end diffusion-based framework for music generation conditioned on dance motion. The music audio is first converted into Mel-spectrograms and then is mapped to the compact latent space $Z_0$. Within this latent space, we perform diffusion processes by progressively adding noise during the forward process and subsequently applying the reverse denoising steps to iteratively recover the data distribution. Formally, by gradually adding Gaussian noise $\epsilon \sim \mathcal{N}(0, I)$ to $Z_0$ according to a variance schedule $\{\beta_1, \cdots, \beta_T\}$, we have

$$q(Z_t|Z_{t-1}) = \mathcal{N}(Z_t; \sqrt{1 - \beta_t} Z_{t-1}, \beta_t I),$$
$$q(Z_{1:T}|Z_0) = \prod_{t=1}^{T} q(Z_t|Z_{t-1}). \tag{9}$$

Then, to recover $Z_0$ from a probability density $p(Z_T)$, the random noise is iteratively denoised through a fixed Markov Chain of length $T$ by a sequence of denoising autoencoders $\theta$ in the reverse process. That is,

$$p(Z_{t-1}|Z_t) = \mathcal{N}(Z_{t-1}; \mu_\theta(Z_t, t), \Sigma_\theta(Z_t, t)),$$
$$p(Z_{0:T}) = p(Z_T) \prod_{t=1}^{T} p_\theta(Z_{t-1}|Z_t), \tag{10}$$

where $\mu_\theta$ denotes the Gaussian mean value.

## B  USER STUDY

Similar to (Sun et al., 2025), we conducted a user study focusing on both music quality and dance-music synchronization. Specifically, We invited 10 participants to evaluate the generated samples, where each participant was randomly presented with 10 generated music clips from the AIST++ dataset, produced by three different methods LORIS, PN-Diffuison and PACE. or each clip, participants were asked to rate on a five-point Likert scale two aspects: (1) the overall quality of the music (OVL), and (2) the relevance between the music and the dance video (REL). The average results are summarized in **Tab. A1**. s shown, our PACE method obtains the highest scores on both metrics, with an OVL of 4.2 and a REL of 4.5, which are substantially higher than those of baseline methods, indicating that PACE consistently produces music that better matches the temporal and stylistic patterns of the dance, confirming the effectiveness of our hierarchical slow-fast motion conditioning.

| Method | AIST++ | |
|---|---|---|
| | OVL↑ | REL↑ |
| LORIS | 2.5 | 2.8 |
| PN-Diffusion | 3.0 | 3.1 |
| **PACE (Ours)** | **4.2** | **4.5** |

Table A1: Mean opinion score on AIST++.

## C  LLM USAGE

We used a large language models (LLMs) solely for minor editing purposes, including grammar checking, wording refinement, and clarity improvement. The LLMs were not involved in research ideation, experimental design, analysis, or result interpretation.

