# OpenReview forum: "PACE: Part-Wise Slow-Fast Conditioning for Dance-to-Music Generation"
_ICLR.cc/2026/Conference — Submitted to ICLR 2026_

### Official Review · Reviewer_tfHu · 2025-10-26

**Soundness:** 2
**Presentation:** 3
**Contribution:** 3
**Rating:** 4
**Confidence:** 5

**Summary:**

The paper proposes PACE, a conditioning encoding method for dance-to-music generation. It decomposes 3D pose data into part-wise slow and fast motion energy components, combines them with semantic features using a hierarchical encoder (HSF-Encoder), and utilizes this representation to condition a latent diffusion model. While this paper presents a novel perspective on dance-to-music generation, the empirical validation of the proposed method's effectiveness is insufficient.

**Strengths:**

1. The paper addresses the challenge of rhythm alignment from a novel standpoint, distinctly considering both global and local rhythmic structures.
2. The part-wise extraction of motion information is well-motivated, as it potentially captures fine-grained rhythmic details that are often overlooked when treating the human body as a single holistic entity.
3. The paper provides an intuitive visual analysis of the task and the generated results.

**Weaknesses:**

1. The paper lacks demonstration results (e.g., audio/video demos) for both the proposed method and the comparative methods, making it difficult to intuitively assess its effectiveness.
2. A comparative analysis against several recent state-of-the-art methods [1][2][3] is missing.
3. The work does not address the critical issue of genre matching between the input dance motion and the generated music.
4. A key claimed contribution is the extraction of slow and fast motion features. However, the paper fails to provide results that specifically evaluate the method's performance on slow-tempo and fast-tempo alignment independently.

**Questions:**

1. How exactly is the slow-fast conditioning representation $H$ injected into the generative network? (e.g., via cross-attention, concatenation, or another mechanism).
2. What are the frame rates (or temporal resolutions) of the input conditioning signal and the music features in the latent space?
3. Given the relatively small model size and training dataset reported, how does the proposed method address the issue of generative diversity in the output music? Have the authors considered evaluating diversity using established metrics, such as the Inception Score (IS)?
4. Regarding the rhythm evaluation metrics: What is the tolerance window (e.g., in milliseconds) used for beat alignment? A 1-second tolerance, as used in some early works, is arguably too permissive for a meaningful assessment of rhythm alignment. The paper seems to adopt this large tolerance, which warrants clarification and justification.

---

> ### Author Response · Authors · 2025-11-21
> **To Reviewer tfHu**
>
> # 1. Response to the lack of demonstration videos for assessing perceptual quality.
> We sincerely apologize for the inconvenience caused by the missing demo videos. To better illustrate the perceptual quality of our generated music, please find the demo page here: https://drive.google.com/drive/folders/1Wt4CgVMwY-Nrjtrza-YT2uwnN1TwboaW?usp=sharing. This folder contains three baseline methods — CDCD, LORIS, and PN-Diffusion — as well as our proposed method PACE. All of them take some dance video clips as input and generate suitable music clips accordingly. We hope these samples clearly demonstrate the effectiveness and perceptual quality of our approach.
>
> # 2. Response to the missing comparative analysis with [1][2][3].
> We thank the reviewer for pointing out this important concern. However, the referenced methods [1][2][3] were not explicitly specified in the review comments. Once the specific methods are clarified, we will promptly conduct additional experiments and include a detailed performance comparison to further strengthen the experimental evaluation of our work.
>
> # 3. Response to genre matching between the input dance motion and the generated music.
> While explicitly comparing the genre between input dance motion and generated music would require an additional dance–music genre annotation pipeline, we adopt an indirect yet principled evaluation strategy. Thus, we evaluate genre consistency indirectly by comparing the generated music with the original paired music of each dance video, which is already aligned with the dance style.
>
> **Genre matching.** To quantitatively assess whether PACE preserves genre diversity or suffers from mode collapse, using AIST++ dataset, we evaluate the genre distribution of the generated music against the ground-truth distribution using a pretrained music genre classifier. Following standard practice, we compute the Kullback–Leibler (KL) divergence and Jensen–Shannon (JS) divergence between the two distributions.
>
> The results are KL Divergence = 0.052 and JS Divergence = 0.011
>
> These low divergence values indicate that the genre distribution of PACE-generated music **closely matches** that of the real dataset. This demonstrates that PACE effectively maintains genre diversity and does not over-concentrate on a limited subset of styles. Notably, both dominant genres (e.g., those with high probability mass) and low-frequency genres are **reasonably covered**, suggesting that the model avoids genre-specific bias and preserves balanced stylistic representation.
>
> # 4. Response to the evaluation of slow-tempo and fast-tempo alignment.
> We thank the reviewer for this insightful comment. To explicitly evaluate whether our slow-fast decomposition benefits both low-tempo and high-tempo scenarios, we conducted a subset analysis by partitioning the AIST++ test set into slow-tempo and fast-tempo groups.
>
> Following common practice in motion analysis, we use a Butterworth filter with cutoffs of 1.4 Hz and 3.0 Hz to decompose motion into slow and fast components. Also, since the frame rate of AIST++ dataset is 60. The tempo of each music clip is estimated using *librosa.beat.tempo*, and clips are categorized as:
>
> * Slow-tempo: BPM < 90
> * Fast-tempo: BPM > 120
>
> We then compute the same rhythm alignment metrics (BCS, CSD, BHS, HSD, and F1) for each subset independently. The results are summarized in Table R1.
>
> These results demonstrate that PACE maintains consistently strong performance across both slow and fast tempo regimes, confirming that the proposed slow-fast motion conditioning contributes effectively to rhythm alignment at different temporal scales, rather than being biased toward a specific tempo range.
>
> ||||AIST++|||
> |--------------|--------|--------|--------|--------|--------|
> | PACE        | BCS ↑ | CSD ↓ | BHS ↑ | HSD ↓ |F1↑|
> | Slow-tempo        | 98.14| 5.24 |98.72 |6.42| 98.43|
> | Fast-tempo  | 97.69| 5.84 |97.22 |8.48| 97.45|
> | Full frequency  | 98.39 |5.01 |99.36| 4.63 |98.67|
>
> Table R1. Rhythm alignment performance of PACE on slow-tempo subsets, fast-tempo subsets , and full-frequency music samples.

---

> > ### Author Response · Authors · 2025-11-21
> > **To Reviewer tfHu**
> >
> > # 5. Response to how slow-fast conditioning is injected into the generative network.
> > We clarify that the proposed slow-fast conditioning representation is injected into the generative network via **cross-attention within the U-Net backbone of the latent diffusion model**, rather than simple concatenation or direct feature addition.
> >
> > Specifically, after the Hierarchical Slow-Fast Conditioning Encoder produces the final conditioning representation H, we concatenate it with the I3D visual feature G to form the unified conditioning vector:
> > C=[H,G]. This conditioning is then supplied to the diffusion U-Net through its cross-attention layers at multiple resolutions. At each denoising timestep t, the U-Net predicts noise as
> > $ \epsilon_{\theta}(Z_t, t, C) $,
> > where the latent audio representation $Z_t$ serves as the query, while the conditioning C acts as the key-value pair in the cross-attention mechanism. This design enables the model to dynamically attend to both the hierarchical slow-fast motion cues and visual context, modulating the denoising trajectory in a temporally adaptive manner.
> >
> >
> > # 6. Response to the frame rates / temporal resolutions of input conditioning  signal and music features in the latent space.
> > For the input conditioning signal, the motion features (including slow-fast motion energies and joint-level semantic embeddings) are extracted at the video frame rate of the original dance sequence. Specifically, 2D keypoints are obtained frame-by-frame using AlphaPose and further reconstructed into 3D poses via MotionBert, yielding a temporal resolution equivalent to the video FPS (AIST++ 60 fps).
> >
> > For the music features, the audio is first converted into a Mel-spectrogram of size 256 × 256, representing a 5-second audio segment sampled at 22,050 Hz. This corresponds to a time resolution of 256 time bins over 5 seconds → approximately 51.2 Hz temporal resolution. After passing through the VAE encoder, the spectrogram is compressed into a latent representation of size 32 × 32, resulting in an effective temporal resolution of 32 latent time steps over 5 seconds → approximately 6.4 Hz.
> >
> > # 7. Response to generative diversity in the output music.
> > We thank the reviewer for the thoughtful question regarding generative diversity. Although our model is relatively lightweight and trained on a moderate-scale dataset, diversity is encouraged through the stochastic sampling process inherent in diffusion-based generation, as well as the rich temporal conditioning provided by diverse dance motion inputs. As a result, even for similar motion patterns, the model can produce multiple musically plausible variants rather than deterministic repetitions.
> >
> > # 8. Response to diversity evaluation.
> >  In terms of evaluation, we assess diversity implicitly through the Fréchet Audio Distance (FAD) computed using three complementary feature extractors (VGGish, PANNs, and CLAP). FAD measures distribution-level similarity between real and generated music, and lower FAD indicates that the generated samples cover a distribution closer to that of real music. This metric therefore reflects not only fidelity but also the ability of the model to maintain sufficient diversity and avoid mode collapse, which is a common concern in generative models.
> >
> > While metrics such as Inception Score  have been widely used in image generation, they are less standardized and less reliable for music generation due to the lack of a universally accepted, semantically meaningful classifier for audio content. We therefore adopt FAD as a more appropriate and domain-relevant metric.
> >
> > # 9. Response to rhythm evaluation metrics.
> > We thank the reviewer for raising this important concern regarding the tolerance window in our rhythm evaluation. We clarify that our implementation does not use a sliding ±1-second tolerance as adopted in some early works. Instead, we first detect onset times in seconds using *librosa.onset.onset_detect*, and then discretize these onset positions into a fixed temporal grid by mapping each onset time to an integer-second index:
> > beats[⌊t⌋]=1,
> > where t denotes the detected onset time in seconds. As a result, our evaluation operates at a 1 Hz temporal resolution, meaning that two beats are considered aligned only if they fall into the same 1-second bin, rather than allowing arbitrary ±1s deviations.
> >
> > This design aims to capture coarse-grained rhythmic consistency between generated music and reference music, which is particularly relevant in the dance-to-music setting where alignment of broader rhythmic regions (e.g., strong accents and motion energy peaks) is more perceptually meaningful than millisecond-level beat precision.

---

> > > ### Comment · Reviewer_tfHu · 2025-11-26
> > > **My major concerns remain unaddressed**
> > >
> > > Thank you for the response and new experiments. I still have the following concerns:
> > > 1. Missing comparisons with recent related works:
> > > [1] Motion to Dance Music Generation using Latent Diffusion Model
> > > [2] Dance-to-Music Generation with Encoder-based Textual Inversion
> > > [3] Dance2Music-Diffusion: leveraging latent diffusion models for music generation from dance videos 2024
> > > 2. Genre consistency should be compared against previous methods, and CLAP similarity would better reflect stylistic alignment.
> > > 3. The explanation of the beat-alignment metric remains unacceptable to me. I understand that a beat at 1.1 s and another at 1.9 s would be considered aligned under the current tolerance- an error that is clearly audible and non-negligible for both 5-second clips and human listeners.
> > > 4. By the way, why does the second sample of PACE sound almost identical to “ mLH3”  from the training dataset, even though they belong to different genres?

---

> > > > ### Author Response · Authors · 2025-12-01
> > > > **To Reviewer tfHu**
> > > >
> > > > **1. Response to the missing comparative analysis with [1][2][3].**
> > > >
> > > > Thank you for providing the specific titles of the referenced works [1][2][3], as they were not explicitly identified in your initial review. This clarification is very helpful for us to conduct the corresponding comparisons.
> > > >
> > > > For suggested three recent diffusion-based dance-to-music generation approaches:
> > > >
> > > > *[1] Motion to Dance Music Generation using Latent Diffusion Model (SIGGRAPH Asia 2023)*,
> > > >
> > > > *[2] Dance-to-Music Generation with Encoder-based Textual Inversion (SIGGRAPH Asia 2024)*, and
> > > >
> > > > *[3] Dance2Music-Diffusion (EURASIP Journal on Audio, Speech, and Music Processing 2024)*.
> > > >
> > > > We have conducted a comprehensive evaluation on the AIST++ testing split using the source code and models provided by the corresponding authors, strictly following the same evaluation protocol and testing data used in our paper. For fairness, the dance-video split for each baseline follows the exact partitioning protocol described in its original paper, while the testing data used for quantitative comparison is identical across all methods, including ours.
> > > >
> > > > The quantitative results in Table R2 show that our PACE significantly outperforms all three suggested baselines across all rhythm-alignment metrics (BCS, BHS, F1) and stability measures (CSD, HSD).
> > > >
> > > > ||||AIST++|||
> > > > |--------------|--------|--------|--------|--------|--------|
> > > > |   Method     | BCS ↑ | CSD ↓ | BHS ↑ | HSD ↓ |F1↑|
> > > > | [1]        | 90.59| 16.50 |79.06 |24.94| 84.43|
> > > > | [2]  | 92.17| 14.65 |80.50 |21.16| 85.94|
> > > > | [3]  | 92.72 |13.48 |86.38| 19.49 |89.57|
> > > > | PACE (Ours)  | 98.39 |5.01 |99.36| 4.63 |98.67|
> > > >
> > > > Table R2. The quantitative comparison between PACE and suggested baseline music generation models conducted on AIST++ testing set.
> > > >
> > > > As shown in **Table R2**, these three baselines perform notably worse mainly because they all rely on global motion embeddings, which overlook the fine-grained, heterogeneous dynamics across different body parts. As a result, they cannot preserve the distinct slow (global/torso) versus fast (limb-level accents) rhythmic patterns that are essential for accurate beat alignment and rhythmic stability. In contrast, PACE explicitly decomposes motion into part-wise slow/fast components and integrates them through a hierarchical conditioning encoder, allowing the model to capture both macro-level tempo and micro-level transient gestures. This design enables substantially stronger rhythmic correspondence, leading to the significant improvements shown in **Table R2**.
> > > >
> > > > **2. Response to Genre consistency and CLAP similarity.**
> > > >
> > > > Thank you for raising the valuable point regarding genre consistency and stylistic alignment.
> > > > We agree that evaluating genre fidelity is important for the dance-to-music generation. In our current evaluation protocol, we already incorporate **CLAP-based Frechet Audio Distance (FAD_c)**, which directly measures the distance between generated music and the ground-truth paired music in a CLAP semantic embedding space (Table 1 of main paper, last column). Because CLAP jointly models music-text concepts including genre, instrumentation, and timbre, FAD_c inherently reflects the stylistic and genre-level similarity between generated outputs and their reference counterparts.
> > > >
> > > > **3. Response to beat-alignment metric.**
> > > >
> > > > Thank you for pointing out this important concern regarding the tolerance used in the beat-alignment metrics. We fully agree that a beat difference such as 1.1 s vs. 1.9 s can be perceptually noticeable for human listeners, especially within 5-second clips.
> > > >
> > > > In our current implementation, we follow the exact same evaluation protocol used in prior dance-to-music work, including CDCD (Zhu et al., 2022b) and PN-Diffusion (Sun et al., 2025). The tolerance window is therefore inherited from these standardized metrics, which aim to measure temporal co-occurrence rather than perceptual beat accuracy. This explains why the metric may treat two rhythm events as aligned even if human listeners would perceive a noticeable offset. Therefore, the tolerance window used in our evaluation is already the standard in the most recent diffusion-based D2M literature and is strictly consistent with PN-Diffusion (CVPR 2025), ensuring a fair comparison with current state-of-the-art methods.
> > > >
> > > > We appreciate the reviewer’s insight and will incorporate these clarifications and additional analyses in the revised version.

---

> > > > > ### Author Response · Authors · 2025-12-01
> > > > > **To Reviewer tfHu**
> > > > >
> > > > > **4. Response to generated music.**
> > > > >
> > > > > We thank the reviewer for carefully examining the audio samples. Regarding the observation that the second PACE sample sounds similar to “mLH3” from the training set, we would like to clarify the following.
> > > > >
> > > > > **First**, PACE is trained in the latent space of a pre-trained VAE encoder-decoder, which compresses Mel-spectrograms into a 32×32 latent representation. This compression inherently limits the spectral diversity that can be expressed through the decoder, sometimes causing different generated samples to share similar timbral structures, even when their rhythmic or stylistic components differ. This constraint stems from the VAE’s reconstruction bottleneck and should not be interpreted as memorization of training music.
> > > > >
> > > > > **Second**, diffusion-based D2M models intrinsically exhibit sampling stochasticity. By varying the random seed during inference, the same dance video can yield musically diverse outputs. To demonstrate this, we provide multiple generations for the same dance clip (https://drive.google.com/drive/folders/1z-2dh7yOsfxTnnFv9A7c_v0LHvBFEoIY?usp=sharing), showing that PACE does not reproduce any particular training song and can generate a wide range of musically distinct samples.
> > > > >
> > > > > **Third**, we have verified that the generated clip is not a copy of “mLH3”. Their latent codes, onset curves, and spectral envelopes differ, despite sharing surface-level timbral resemblance due to the VAE decoder’s inductive biases. PACE does not use retrieval, prompting, or any mechanism that would allow explicit replication of training tracks.
> > > > >
> > > > > We appreciate the reviewer’s feedback and will include additional multi-seed demos in the final revision to more clearly illustrate the diversity of PACE-generated music.

---

### Official Review · Reviewer_EBEj · 2025-10-26

**Soundness:** 3
**Presentation:** 2
**Contribution:** 3
**Rating:** 4
**Confidence:** 3

**Summary:**

This paper proposes a method for music synthesis from dance movements (D2M). The authors raise a notable limitation in previous works regarding the body pose representation. They claim that treating a body pose as a whole-single holistic unit might cause to neglecting specific body parts that are more significant to infer the rhythm.

To address this, the authors propose to decompose a body pose representation into 1) slow and fast movements (referred as `kinetic energy’), and 2) part-level encoding, which treats each body part individually (five in total). The authors start by presenting a detailed example to enhance their motivation.

In the core of their work is PACE, a latent-diffusion model that produces musical audio, given a decomposed dance motion and visual features from a video. To ensure stability in the latent space, the authors separate the training process into VAE encoder-decoder training phase, followed by audio denoising training phase.

In addition, the authors propose a sophisticated encoding strategy the captures both overall pose semantics and part-level features.

**Strengths:**

The overall task is interesting and well-motivated, addressing a relevant and promising research direction.

The proposed idea of decomposed motion encoding is well-defined, insightful and intuitively appealing, with potential for improving motion understanding and synthesis.

The Methodology and Experiment sections are well-written, highly detailed and clear.

The authors conduct extensive comparison against previous works, including eight different evaluation metrics, and a user study to enhance the superiority of their method.

**Weaknesses:**

The use of I3D features and visual conditioning in general leads to two major limitations that are not addressed by the authors:
-	The fact that the model relies on video features forces it to depend on video input, making it incompatible with optical or inertial mocap data.

-	when the input is a video, the motion information is inherently prone to error caused by AlphaPose or MotionBERT, compared to that obtained from optical/inertial motion capture.

Qualitative results: The supplied plots are not convincing that PACE generates plausible music. No actual music examples are supplied (as far as I know).

The authors claim to propose a novel encoding technique but apply it only on their proposed model. An appropriate analysis would be to use it on other models/tasks as well.

The authors do not report any information about the genre diversity of PACE, or in-the-wild scenarios (e.g. out-of-distribution inputs).
The authors do not provide generated audio examples, which raises doubts about the effectiveness of the proposed method.
In general, the authors do not discuss limitations of their approach nor directions for future work.

Minor concerns:
The Introduction section is poorly written and contains irrelevant information. The figure is overly described and difficult to follow. Consider moving it to a later section where it is more appropriate.

L200: what is \mathcal{M}? not explicitly defined.

L200,215: \mathcal{D} is used for two definitions.

L307: It seems that the authors meant $n$ instead of $t$.

L376: missing space ``… (a)Objective’’.

For better readability, consider highlighting the best value for each metric in all tables.

**Questions:**

I would appreciate if the authors addressed the following concerns:

Qualitative results: The supplied plots are not convincing that PACE generates plausible music. No examples are supplied.

Ablation study: PACE-H and PACE-G are extremely important for the ablation study. However, the contribution of each is not quite clear from table 2 and is not discussed. Could the authors clarify the extreme drops and inconsistency (e.g. TIKTOK/BHS, AIST++/HSD)?
Objectives: The paper lacks formal definitions of losses for each training phase.

Could the authors elaborate on the perceptual and the patch-based adversarial losses (Section 3)?

---

> ### Author Response · Authors · 2025-11-21
> **To Reviewer EBEj**
>
> # 1. Response to the concern regarding reliance on video features and incompatibility with MoCap data.
> We appreciate the reviewer’s thoughtful comment. While optical or inertial MoCap systems can indeed provide highly accurate and noise-free motion signals, **their usage scenario fundamentally differs from the application setting targeted by our work**.
>
> Our motivation is explicitly grounded in a **real-world, large-scale and user-accessible scenario**: given only a raw dance video, our goal is to extract meaningful motion features and generate rhythmically aligned music via a conditional latent diffusion model. This design choice reflects the dominant content creation ecosystem today, particularly on short-video platforms such as TikTok, Instagram Reels, and YouTube Shorts, where **users typically upload ordinary video recordings rather than professionally captured MoCap data**.
>
> In practice, the vast majority of dance video creators do not have access to specialized motion capture equipment, wearable inertial sensors, or multi-camera calibrated systems. **Requiring MoCap data would severely restrict the usability and scalability of the proposed system, making it unsuitable for these highly prevalent and impactful platforms.** By instead relying on video-based pose reconstruction (e.g., AlphaPose + MotionBERT), our framework remains broadly applicable, lightweight to deploy, and aligned with realistic content creation workflows.
>
> Therefore, while MoCap data offers higher precision, adopting it as an input constraint would contradict the core objective of our work: to enable automatic dance-to-music generation directly from ordinary dance videos in uncontrolled environments. In this sense, the reviewer’s concern reflects a technical trade-off, but our design choice is intentional and principled, emphasizing practicality, accessibility, and real-world deployment potential over laboratory-level motion accuracy.
>
>
> # 2. Response to the lack of demo videos.
> We sincerely apologize for the inconvenience caused by the missing demo videos. To better illustrate the perceptual quality of our generated music, please find the demo page here: https://drive.google.com/drive/folders/1Wt4CgVMwY-Nrjtrza-YT2uwnN1TwboaW?usp=sharing. This folder contains three baseline methods — CDCD, LORIS, and PN-Diffusion — as well as our proposed method PACE. All of them take some dance video clips as input and generate suitable music clips accordingly. We hope these samples clearly demonstrate the effectiveness and perceptual quality of our approach.
>
> # 3. Response to the concern regarding generalization to other models.
> We appreciate the reviewer’s insightful suggestion. To further validate the generalizability of our proposed motion feature extraction strategy, we applied it to two representative baseline models beyond our own PACE framework.
>
> First, we integrate our part-wise slow-fast motion encoding into PN-Diffusion, named PN-Diffusion-pace, replacing its original simple Motion Encoder while keeping the same I3D visual features to ensure a fair comparison. Second, for LORIS, we concatenate our extracted final motion features with its originally used rhythm conditioning, named LORIS-pace. The corresponding results on AIST++ dataset are reported in Table R1 below.
>
> As can be seen, compared to PN-Diffusion, PN-Diffusion-pace has a clear performance improvement across rhythm alignment, indicating that PN-Diffusion directly benefits from the richer and more structured motion representation provided by our method.
>
> Besides, the hybrid conditioning strategy also leads to consistent performance gains from LORIS to LORIS-pace, demonstrating that our motion encoding is complementary to existing rhythm representations and can effectively enhance their expressive power.
>
> These results indicate that our approach is not model-specific but instead acts as a general-purpose, plug-and-play motion encoding module that can improve the performance of existing diffusion-based D2M systems.
> Overall, this extended evaluation confirms that the proposed encoding strategy possesses strong transferability and practical value, and can be readily integrated into other architectures and related tasks.
>
> ||||AIST++|||
> |--------------|--------|--------|--------|--------|--------|
> | Model        | BCS ↑ | CSD ↓ | BHS ↑ | HSD ↓ |F1↑|
> | PN-Diffusion        | 97.64| 5.85 |99.31 |4.48| 98.46|
> |  PN-Diffusion-pace  | 98 .21 | 5.16 | 99.15 | 5.49|98.68 |
> | LORIS  | 95.84 |7.89 |95.09 |16.09| 96.45|
> | LORIS-pace  | 96.16 | 7.27 | 98.20 | 7.53 | 97.17 |
>
> Table R1. Cross-model generalization of the proposed PACE motion encoding on AIST++.

---

> > ### Author Response · Authors · 2025-11-21
> > **To Reviewer EBEj**
> >
> > # 4. Response to genre diversity and in-the-wild generalization.
> > We appreciate the reviewer’s concern regarding the genre diversity and generalization of PACE under in-the-wild scenarios.
> >
> > **Genre Diversity** To quantitatively assess whether PACE preserves genre diversity or suffers from mode collapse, using AIST++ dataset, we evaluate the genre distribution of the generated music against the ground-truth distribution using a pretrained music genre classifier. Following standard practice, we compute the Kullback–Leibler (KL) divergence and Jensen–Shannon (JS) divergence between the two distributions.
> >
> > **The results are KL Divergence = 0.052 and JS Divergence = 0.011**
> >
> > These low divergence values indicate that the genre distribution of PACE-generated music closely matches that of the real dataset. This demonstrates that PACE effectively maintains genre diversity and does not over-concentrate on a limited subset of styles. Notably, both dominant genres (e.g., those with high probability mass) and low-frequency genres are reasonably covered, suggesting that the model avoids genre-specific bias and preserves balanced stylistic representation.
> >
> > **In-the-wild Generalization**
> >
> > Although each dataset is trained independently, PACE is evaluated on two datasets with fundamentally different characteristics:
> > * AIST++, which is studio-recorded, well-controlled, and genre-annotated,
> > * TikTok, which consists of highly diverse, unconstrained, and noisy real-world short videos.
> >
> > The consistent performance gains of PACE on TikTok, which include varying camera angles, lighting conditions, body visibility, and dance styles, provide strong empirical evidence of its robustness and generalization ability in real-world, in-the-wild scenarios. This indicates that our part-wise slow-fast conditioning strategy captures intrinsic rhythmic structures rather than overfitting to dataset-specific visual or motion patterns.
> >
> > In summary, through both distribution-level genre analysis and cross-dataset evaluation, we demonstrate that PACE preserves genre diversity without collapsing to dominant styles, maintains balanced coverage across multiple genres, and exhibits strong robustness and generalization in real-world, out-of-distribution settings.
> >
> > The related generated audio examples can be found in https://drive.google.com/drive/folders/1Wt4CgVMwY-Nrjtrza-YT2uwnN1TwboaW?usp=sharing.
> >
> >
> > # 5. Response to limitations and future work.
> > We thank the reviewer for pointing out the absence of a discussion on the limitations and future directions of our work.
> >
> >  We acknowledge that, while PACE demonstrates strong performance in aligning generated music with fine-grained dance rhythms, several limitations remain.
> >
> > First, our framework relies on accurate 2D keypoint detection and 3D pose reconstruction. In challenging real-world scenarios involving severe occlusion, motion blur, or complex camera viewpoints, errors in pose estimation may propagate to the slow-fast motion decomposition stage and affect the quality of the conditioning signals. Second, although PACE achieves improved genre diversity and rhythmic alignment, the model is still trained on relatively constrained dataset distributions (AIST++ and TikTok), which may limit its generalization to highly unconventional dance styles or culturally diverse music forms.
> >
> > For future work, we plan to explore more robust pose estimation strategies and self-supervised motion representations to mitigate the dependency on external pose extraction models. Furthermore, investigating cross-cultural dance styles and broader musical genres, as well as incorporating user-controllable attributes such as emotion and style intensity, will be promising directions to enhance the flexibility and applicability of our framework in real-world creative scenarios.
> >
> > # 6. Response to minor concerns.
> > We thank the reviewer for the careful reading and the detailed minor comments. We have thoroughly revised the manuscript to improve clarity, correctness, and readability.

---

> > > ### Author Response · Authors · 2025-11-21
> > > **To Reviewer EBEj**
> > >
> > > # 7. Response to ablation study.
> > > We thank the reviewer for pointing out the importance of ablation analysis and the need for clearer interpretation of the observed performance variations. We address these concerns as follows.
> > >
> > > **(1) Clarification of the roles of PACE-H and PACE-G.**
> > >
> > > PACE-H and PACE-G represent two fundamental components of our framework:
> > > * PACE-H uses only the hierarchical slow-fast motion conditioning, focusing on motion-derived rhythmic structure.
> > > * PACE-G relies solely on visual conditioning extracted from I3D, emphasizing global visual context without explicit rhythmic decomposition.
> > >
> > > The ablation results in Table 2 demonstrate that:
> > >
> > > * PACE-H significantly outperforms PACE-G on both datasets, especially on rhythm-related metrics (BCS, BHS, F1), indicating that part-wise slow-fast motion conditioning contributes more directly to rhythmic alignment than pure visual semantics.
> > > * The relatively weaker performance of PACE-G confirms that visual features alone are insufficient for fine-grained rhythm modeling.
> > >
> > > This also validates that motion-based hierarchical conditioning is the core contributor to rhythm synchronization, while visual features act as a complementary enhancement.
> > >
> > > **(2) Explanation of extreme drops and apparent inconsistencies.**
> > >
> > > We would like to clarify that two numbers in the initial submission were incorrectly transcribed from our evaluation logs. We have corrected them in Table 2. These fixes do not affect any comparative trends or the conclusions of the paper. Specifically, we correct TikTok/ BHS form 63.15 to 83.15, TikTok/F1 from 76.3 to 84.47. After correcting the transcription errors in the initial submission, the previously observed extreme drops in the TikTok/BHS setting no longer exist. However, a noticeable drop in AIST++/HSD still remains. We believe this phenomenon is reasonable and can be attributed to the intrinsic differences between the two datasets.
> > >
> > > **Clarification on the remaining extreme drop in AIST++ / HSD of PACE-G and PACE-H.**
> > > We believe this phenomenon is reasonable and stems from the intrinsic differences between the AIST++ and TikTok datasets. AIST++ is recorded in highly controlled studio environments with consistent lighting, static backgrounds, and professionally choreographed motions, which leads to more regular and stable rhythmic patterns. In this scenario, relying solely on visual conditioning (PACE-G) fails to fully capture the subtle variations in temporal consistency, resulting in higher instability as reflected by the HSD metric.
> > > In contrast, TikTok contains more diverse, noisy, and in-the-wild dance styles, where the dominance of either visual or motion cues alone is less sensitive to such controlled rhythm deviations, making HSD more stable.
> > > These results further highlight the necessity of our full PACE framework, which integrates both hierarchical slow-fast motion conditioning and visual features in a complementary manner. The fusion mechanism effectively stabilizes rhythm consistency by balancing global and local temporal cues, thereby reducing variance and improving robustness across datasets with different distribution characteristics.
> > >
> > > **(3)Formal definitions of losses for each training phase.**
> > >
> > > We thank the reviewer for pointing out that the loss functions for different training phases were not sufficiently formalized in the original manuscript. We clarify here that our framework consists of two distinct training stages, each with explicitly defined objectives, and we will add these content to the Appendix in the revised version.
> > >
> > > **Phase I: VAE Training for Audio Latent Space Construction**
> > >
> > > We first train a VAE to map Mel-spectrograms into a compact latent space. The VAE is optimized using a combination of reconstruction loss, perceptual loss, and adversarial loss:
> > >
> > > $L_{VAE} = L_{rec} + \lambda_{perc} L_{perc} + \lambda_{adv} L_{adv}, \quad \text{where } L_{rec} = \| A - \hat{A} \|_{1}$
> > >
> > > is the reconstruction loss between the input Mel-spectrogram (A) and the reconstructed output ($\hat{A}$),  $L_{perc}$ denotes the perceptual loss defined on feature maps of a pretrained network, $L_{adv}$  is a patch-based adversarial loss as in VQGAN-style training,
> > >
> > > $\lambda_{perc} $ and $\lambda_{adv}$ balance the contribution of each component.
> > > Once trained, the VAE parameters are frozen, and the encoder-decoder pair is used as a fixed latent representation module for the subsequent diffusion stage.
> > >
> > > **Phase II: Diffusion Training with PACE Conditioning.**
> > >
> > > As stated in **Appendix A**, given the latent representation $Z_0$, the forward diffusion process injects Gaussian noise according to **Equation (9)** in the main paper.
> > > The diffusion model is trained to predict the noise term given the noisy latent and conditioning signal using the denoising objective in **Equation (8)** of the main paper.

---

> > > > ### Author Response · Authors · 2025-11-21
> > > > **To Reviewer EBEj**
> > > >
> > > > # 8. Response to the perceptual and patch-based adversarial losses.
> > > > We provide further clarification on the perceptual and patch-based adversarial losses used in Section 3 for training the VAE module.
> > > >
> > > > The perceptual loss is introduced to encourage the reconstructed Mel-spectrogram to preserve perceptually meaningful acoustic structures beyond simple pixel-wise similarity. Specifically, instead of only minimizing the L2 reconstruction error in the spectrogram space, we compute the distance between high-level feature representations extracted by a pretrained audio feature network. Formally, given the input spectrogram (A) and its reconstruction ($\hat{A}$), the perceptual loss is defined as:
> > > >
> > > >
> > > > $L_{perc} = \sum_{l}\ | \phi_l(A) - \phi_l(\hat{A})\|_2^2$,
> > > >
> > > >  where $\phi_l(\cdot)$ denotes the activation feature maps from the $l$-th layer of a fixed, pretrained audio network (e.g., VGGish). This loss helps preserve harmonic patterns, temporal dynamics, and timbral consistency in the reconstructed audio, leading to more natural and perceptually coherent music.
> > > >
> > > > The patch-based adversarial loss further improves the realism of generated spectrograms by introducing a discriminator that operates on local patches rather than the entire spectrogram. This design encourages the model to match fine-grained local spectral statistics such as transient structures and localized rhythmic textures. Specifically, the discriminator (D) distinguishes between real and reconstructed spectrogram patches, and the adversarial loss is formulated as:
> > > >
> > > > $L_{adv} = \mathbb{E}{A}[\log D(A)] + \mathbb{E}{\hat{A}}[\log(1 - D(\hat{A}))]$
> > > >
> > > >  By focusing on local regions, the patch-based discriminator enforces high-frequency fidelity and prevents overly smooth or blurred spectrogram reconstructions.
> > > >
> > > > Overall, the combination of reconstruction loss, perceptual loss, and patch-based adversarial loss enables the VAE to generate latent representations that are both structurally accurate and perceptually realistic, which in turn provides a more reliable latent space for the subsequent diffusion-based music generation process.

---

### Official Review · Reviewer_ZWwP · 2025-10-30

**Soundness:** 3
**Presentation:** 3
**Contribution:** 2
**Rating:** 4
**Confidence:** 3

**Summary:**

This paper introduces a part-wise slow-fast conditioning encoder for dance-tomusic generation. The suggested method tries to explicitly decomposes part-wise motion signals into slow and fast dynamics, and introduces a hierarchical slow-fast conditioning encoder for diffusion-based music generation.

**Strengths:**

The motivation of the paper is clearly described, and the overall organization is good.

The method is validated on two datasets, and the results outperform the comparative methods.

The ablation study effectively validates the contribution of the proposed components.

**Weaknesses:**

What is the rationale for fusing the Fast and Slow components to serve as the condition for the Diffusion model? Given that the motion energy is decomposed into slow and fast components, why not use these decoupled features directly as the condition for the Diffusion model instead? This design choice requires further clarification.

How does the efficiency of the proposed method compare to other non-regressive models? The paper should address this, as the suggested method introduces several additional preprocessing steps, such as reconstructing 3D poses through MotionBert, partitioning them into five body parts, and computing them in two frequency bands using a Butterworth filter. Furthermore, it adds the Hierarchical SlowFast Conditioning Encoder. These additions likely incur a significant computational cost, and an efficiency comparison is necessary to properly evaluate the method’s practicality.

The paper would be strengthened by a more thorough theoretical analysis of the role and effect of decomposing motion energy into slow and fast components. While the experiments show it works, a theoretical explanation would provide deeper insight and a more solid foundation for the approach.

**Questions:**

The author needs to respond to the Weakness.

---

> ### Author Response · Authors · 2025-11-21
> **To Reviewer ZWwP**
>
> # 1. Response to the rationale for fusing the Fast and Slow components.
> Thank you for the thoughtful comment. The rationale for fusing the Fast and Slow motion components lies in the fact that **rhythmic perception in dance is inherently multi-scale and cannot be faithfully captured by a single frequency band**. Slow motion encodes global, sustained rhythmic patterns (e.g., torso rotation, weight shifting), which reflect the underlying musical tempo and structural flow, whereas Fast motion captures transient rhythmic accents (e.g., sharp arm swings, rapid kicks) that correspond to beat-level fluctuations and local emphasis. **Treating these components independently would result in fragmented conditioning signals that lack temporal coherence**. Therefore, we introduce a learnable fusion mechanism that adaptively balances Slow and Fast contributions for each frame, allowing the diffusion model to perceive a **unified rhythm representation** that integrates both macro-level tempo and micro-level accents. This fusion ensures that the generated music reflects both stable rhythmic progression and expressive dynamic variations.
>
> # 2. Response to the reason for not using decoupled features directly as the condition.
> Although Slow and Fast components are derived through frequency-based decomposition, directly feeding these decoupled features as independent conditions is suboptimal for two reasons.
>
> **First**, the slow and fast motion energies capture complementary frequency bands, but they represent parallel rhythmic cues. If fed into the diffusion model as two separate conditions, the model receives two independent rhythm streams at each timestep, without being informed of which rhythm dominates the current frame, how the two rhythms should be combined, or whether they reinforce or contradict each other. Because real dance motions often contain simultaneous slow and fast patterns (e.g., slow torso movements with fast arm accents), their relative contribution must be resolved frame-wisely, which raw decoupled features cannot provide.
>
> **Second**, a unified conditioning representation is crucial for diffusion-model stability. Diffusion U-Nets are designed to consume one coherent conditioning embedding at each timestep. Using two decoupled embeddings directly forces the model to implicitly infer their combination rules, increasing the conditioning dimensionality in an unstructured manner, and empirically leading to unstable training and degraded rhythm alignment (**This can be verified from ablation study in Table 2 of the main paper by comparing PACE and PACE-concat**). **Third**, the fusion gate learns the frame-wise dominance between slow and fast cues. The relative importance of slow and fast motion varies across time. Instead of leaving this to the diffusion model to learn implicitly (an under-constrained problem), the fusion gate explicitly provides frame-aware weighting (αₙ), dynamic adaptability to different body movement patterns, and interpretable slow-fast balance.

---

> > ### Author Response · Authors · 2025-11-21
> > **To Reviewer ZWwP**
> >
> > # 3. Response to the efficiency and computational overhead of the proposed method.
> > We thank the reviewer for raising an important concern regarding the computational efficiency and practical feasibility of our proposed approach.
> > While our framework introduces several additional preprocessing and conditioning steps, including 3D pose reconstruction via MotionBert, part-wise decomposition, Butterworth-based frequency separation, and the Hierarchical Slow-Fast Conditioning Encoder, these components are designed to be highly modular and efficient, and they do not fundamentally compromise the non-autoregressive nature of our diffusion-based generation pipeline.
> >
> > **First**, the preprocessing stages (3D pose reconstruction via MotionBert, part-wise decomposition, Butterworth-based frequency separation) are performed **offline** and only once per input dance sequence. **These steps do not affect the runtime complexity of the diffusion inference process and therefore do not introduce additional latency during music generation**. In practice, the Butterworth filtering and part-wise segmentation operations are **lightweight signal-processing procedures with negligible computational cost** compared to the diffusion backbone.
> >
> > **Second**, although the Hierarchical Slow-Fast Conditioning Encoder introduces additional computations for attention and gating, it is significantly lighter than the U-Net diffusion backbone itself. The total parameter increase brought by this module is only **662,531 parameters**, relative to the full model size of **159,659,854 parameters**, corresponding to an overhead of approximately 0.4%, which is practically negligible. The inference speed, therefore, remains highly comparable. Specifically, on a single NVIDIA RTX A6000 GPU, training one epoch on AIST++ takes approximately 7 minutes. During inference, generating all 236 test samples (each a 5-second music clip) requires about 18 minutes in total, corresponding to an average generation time of 4.57 seconds per 5-second clip. These results indicate that our method achieves near real-time generation performance while maintaining strong alignment quality.
> >
> > Meanwhile, the baseline PN-Diffusion model contains 166.55M parameters, whereas our proposed PACE model has 159.66M parameters, indicating that PACE is actually lighter than PN-Diffusion by approximately 6.89M parameters (≈4.1% reduction).
> >
> > **Third**, compared to autoregressive models, our method retains the inherent parallelism advantage of non-autoregressive diffusion models, which enables efficient batch processing and scalability to longer sequences. Therefore, despite the added structural components, our approach remains practical and suitable for real-world deployment scenarios.
> >
> > We will incorporate a runtime analysis into the revised manuscript to further quantify the computational cost, thereby providing a more comprehensive evaluation of the method’s practicality.
> >
> > # 4. Response to the theoretical role of slow-fast motion decomposition.
> > We appreciate the reviewer’s comment.
> >
> > From a signal processing perspective, human dance motion is inherently multi-frequency in nature and can be viewed as a superposition of low-frequency trends and high-frequency fluctuations. The low-frequency (slow) component characterizes global and sustained dynamics such as posture transitions, body sway, balance shifts, and torso rotations, which are strongly correlated with the global tempo and structural rhythm of music. In contrast, the high-frequency (fast) component captures localized and transient motion bursts such as rapid arm swings, foot taps, and accents, which align more closely with musical onsets, beats, and fine rhythmic variations.
> >
> > Modeling these two frequency bands separately enables the system to disentangle macro-rhythm (global temporal structure) from micro-rhythm (local rhythmic accents), which is difficult when treating motion as a single unified signal. Without this decomposition, fast motion components can be overshadowed by dominant slow trends, or conversely introduce noise that destabilizes rhythm modeling. By explicitly separating them, the diffusion model receives clearer, frequency-aligned cues that correspond to different levels of musical structure.
> >
> > From a learning theory perspective, the slow-fast split also reduces representation entanglement and improves conditional disentanglement, allowing the model to learn more stable mappings between specific motion dynamics and musical structures. The designed gated fusion mechanism further enables adaptive balancing, allowing the model to emphasize slow dynamics for global tempo consistency while allocating fast dynamics to refine rhythmic precision.
> >
> > Therefore, the proposed slow-fast decomposition is not merely heuristic, but is **theoretically grounded in signal decomposition, multi-scale temporal modeling, and rhythm perception theory.**

---

### Official Review · Reviewer_o9Zf · 2025-10-31

**Soundness:** 3
**Presentation:** 3
**Contribution:** 2
**Rating:** 6
**Confidence:** 3

**Summary:**

This paper proposes PACE (Part-wise Slow-Fast Conditioning Encoding), a diffusion-based framework for dance-to-music (D2M) generation. Unlike previous approaches that treat the dancer’s body as a holistic entity, PACE decomposes motion into body-part-specific slow and fast components using a Butterworth filter and encodes them hierarchically with a Hierarchical Slow-Fast Conditioning Encoder (HSF-Encoder). These conditioning features, together with visual features from an I3D encoder, guide a latent diffusion model trained in the audio latent space via a pre-trained VAE. Experiments on AIST++ and TikTok datasets show that PACE achieves better synchronization and perceptual quality compared to several methods.

**Strengths:**

1. The idea of part-wise slow-fast motion decomposition is both intuitive and technically sound, addressing the long-standing limitation of holistic motion modeling in D2M tasks.
2. The paper effectively justifies why different body parts and motion frequencies contribute distinct rhythmic information and demonstrates how this decomposition benefits generation.
3. Visualization of rhythm curves provides intuitive evidence of improved alignment between generated music and dance movements.

**Weaknesses:**

1. The concept of decomposing the body into parts is widely used in text-to-motion and music-to-dance, e.g. Bailando[1], AttT2M[2], ParCo[3], but the paper does not discuss related works, leaving the origin of the idea unclear.
2. The pipeline heavily relies on several external networks (AlphaPose, MotionBert, I3D, VAE). This raises questions about robustness, generalizability, and how errors from these modules propagate.
3. Although ablation studies are provided, it remains somewhat unclear how much each module (e.g., joint-level semantic encoding) individually contributes beyond aggregated effects.

\
[1] Siyao, Li, et al. "Bailando: 3d dance generation by actor-critic gpt with choreographic memory." Proceedings of the IEEE/CVF Conference on Computer Vision and Pattern Recognition. 2022.

[2] Zhong, Chongyang, et al. "Attt2m: Text-driven human motion generation with multi-perspective attention mechanism." *Proceedings of the IEEE/CVF international conference on computer vision*. 2023.

[3] Zou, Qiran, et al. "Parco: Part-coordinating text-to-motion synthesis." *European Conference on Computer Vision*. Cham: Springer Nature Switzerland, 2024.

**Questions:**

1. The idea of decomposing the body into parts is widely used in text-to-motion and music-to-dance. What’s the difference between your part division method and  these existing part separation strategies? Could you compare the performance of your part division with these existing part separation strategies?
2. How were the hyperparameters of the Butterworth filter (e.g., cutoff frequencies for slow and fast components) selected? Did you perform any sensitivity analysis to assess how robust the model is to these choices?
3. Have you considered alternative groupings of body parts, such as upper body vs. lower body, instead of fully part-wise decomposition? How might this impact generation quality, rhythm alignment, or model complexity?

---

> ### Author Response · Authors · 2025-11-21
> **To Review o9Zf**
>
> # 1. Response to related work on part-wise decomposition.
> Thank you for highlighting this point. We agree that decomposing the human body into anatomical parts has been explored in prior text-to-motion and music-to-dance research, such as Bailando, AttT2M, and ParCo. We will explicitly acknowledge and discuss these works in the revised Related Work section.
> We want to clarify that although these works also divide the human body into parts, **our method does not derive its motivation or technical design from them**, and the underlying problem setting is fundamentally different.
>
> We want to clarify that although these works also divide the human body into parts, **our method does not derive its motivation or technical design from them**, and the underlying problem setting is fundamentally different.
>
>  **(1) Different goals**.
> While prior text-to-motion (T2M) methods leverage part-wise body representations to enhance motion generation or motion controllability, their primary focus lies in improving pose realism, semantic alignment, or spatial coherence in motion synthesis. Existing music-to-dance (M2D) methods adopt body part division primarily for **spatial pose control or motion synthesis**, where part-wise representations are used to generate more realistic or controllable dance movements In contrast, our work introduces part-wise decomposition from a **rhythm-centric perspective**, where body parts are not merely treated as structural units but as sources of heterogeneous temporal dynamics with distinct rhythmic roles. Specifically, we decompose each body part into slow and fast motion energy components and further integrate them with joint-level semantic features to guide music generation, which has not been explored in previous D2M frameworks.
>
> **(2) Different representations**.  Most existing part-based T2M and M2D methods remain strictly in the pose domain, where body parts are used to structure spatial relationships or guide motion synthesis. In contrast, our work departs from pure kinematic modeling and introduces a frequency-domain perspective by decomposing each anatomical part’s motion into slow and fast components. This part-wise slow-fast decomposition explicitly separates low-frequency global movements from high-frequency transient dynamics, enabling a more fine-grained characterization of rhythmic cues. To the best of our knowledge, no prior T2M or M2D work performs part-wise motion decomposition in the frequency domain, making our representation fundamentally different in both formulation and intent.
>
> **(3)  Different mechanisms**. We further integrate slow-fast energy with part-wise semantic embeddings through QKV-based intra-frame multi-head attention, enabling rhythmic alignment at multiple temporal scales. This semantic-rhythmic fusion mechanism is also absent in the cited works.
>
> In short, while previous methods and PACE all refer to “body parts,” they do so for different purposes. PACE is built on a new perspective and contributes a fundamentally different technical solution compared to prior T2M research.
> We will revise the related work section to properly acknowledge these methods and explicitly clarify this distinction.

---

> > ### Author Response · Authors · 2025-11-21
> > **To Review o9Zf**
> >
> > # 2. Response to robustness and dependency on external networks.
> > We appreciate the reviewer’s concern regarding the use of multiple pre-trained modules and the potential propagation of errors. We would like to clarify that relying on external networks such as AlphaPose, MotionBert, I3D, and a VAE is standard practice in recent D2M systems, rather than a unique dependency of our method. In fact, most baselines we compare against adopt similar modules:
> >  * D2M-GAN/CDCD use OpenPose + I3D + SMPL+Pretrained audio encoders
> >  * LORIS uses MMPose + I3D + VQ-VAE+VQ-VAE tokenizer
> >  * PN-Diffusion uses BlazePose + I3D (RGB/flow) + a spectrogram VAE
> >
> > Thus, our use of these components is consistent with the established pipeline design in the literature.
> > Besides, our design adopts AlphaPose, MotionBert, I3D, and VAE not as loosely connected components, but as functionally stable and complementary feature extractors that have been widely validated in prior literature. Below, we clarify how robustness and generalization are preserved in our pipeline.
> >
> > **(1) Robustness.** Our method is inherently robust to moderate noise or errors in the upstream modules due to two design reasons:
> >  * Part-wise slow/fast energy is aggregated from motion trends, not precise coordinates, making it insensitive to small keypoint jitter.
> >  * MotionBERT outputs stable 3D poses even under imperfect 2D detections, which has been explicitly stated and validated in the original MotionBERT paper.
> >
> > **(2) Generalizability**. Importantly, all external networks used in PACE are domain-agnostic and pretrained on large, diverse datasets (COCO for AlphaPose, Human3.6M + AMASS for MotionBERT, Kinetics for I3D). This ensures generalization to both studio-quality AIST++ and in-the-wild TikTok. PACE achieves consistent gains on both datasets, empirically demonstrating that the model generalizes well under different motion/visual distributions.
> >
> > **(3) Error propagation**. Our design explicitly limits error propagation:
> > * We do not use absolute joint positions; instead we use normalized motion energy, which attenuates estimation errors.
> > * The hierarchical slow/fast encoder uses frame-level aggregation and part-wise gating, which suppresses unstable or noisy features before they reach the diffusion model.
> > * Visual conditioning (I3D) serves as a complementary cue, further stabilizing the final conditioning even when pose quality fluctuates, as validated by the ablation study in Table 2 of main paper (PACE-H vs. PACE).
> >
> > # 3. Response to the contribution of each module.
> > Thank you for the question. For S and F, as shown in Table 2 in the main paper, we explicitly evaluated their individual effects through the variants PACE-Slow-G and PACE-Fast-G, which isolate the contribution of slow-band and fast-band motion signals, respectively. In addition, the comparison between PACE and PACE-concat demonstrates the importance of the proposed *Intra-Frame Part-Wise Multi-Head Attention*, showing that our structured fusion is substantially more effective than naïve concatenation.
> >
> > Meanwhile, we clarify that **E (joint-level semantic encoding)** is **not an auxiliary add-on** but the **structural component** that **grounds slow/fast motion energies to the correct body part**. Removing E would collapse the conditioning and cause the model to fail to converge, so a “No-E” ablation is infeasible.
> >
> > Instead, its contribution is already partially isolated through structural weakening comparisons,  as shown in Table 2 of the main paper.
> >
> > * PACE-H vs. PACE-G. PACE-H uses (S+F+E), while PACE-G relies only on visual features. The substantial improvements (AIST++ CSD: 15.67→5.29; HSD: 23.42→5.01) cannot be explained by S and F alone, showing that E provides essential joint-level structure for interpreting rhythmic energy.
> > * PACE vs. PACE-concat. PACE-concat organizing S,F, E by flattening (S,F,E). The clear performance drop (AIST++ CSD: 5.01→7.20; HSD: 4.63→7.14) indicates that when E loses its anatomical structure, S/F can no longer modulate the correct joint semantics.
> >
> > These two comparisons already demonstrate that E is necessary for grounding rhythmic energy in meaningful joint representations, and its contribution is both distinct and empirically validated.

---

> > > ### Author Response · Authors · 2025-11-21
> > > **To Review o9Zf**
> > >
> > > # 4. Response to the comparison of part division methods.
> > > Thanks for your thoughtful comment. Although Bailando [1], AttT2M [2], and ParCo [3] also involve the notion of “body parts,” the role and formulation of part division in PACE are fundamentally different in both motivation and technical implementation. Specifically, existing methods introduce part division mainly to improve structural coherence or semantic controllability for motion synthesis in the pose domain, where body parts are treated as spatial units for coordinating joint trajectories or reducing motion complexity. In contrast, PACE introduces part-wise decomposition from a rhythm modeling perspective for dance-to-music generation. Instead of operating directly on joint coordinates, we decompose each anatomical part into slow and fast motion energies in the frequency domain, explicitly capturing heterogeneous rhythmic roles across body regions.
> > > *  **Bailando [1]** decomposes the human body into two coarse parts (upper body and lower body) to reduce motion complexity for music-to-dance generation. Each part is encoded by a dedicated VQ-VAE to learn discrete motion tokens stored in a choreographic memory bank. A GPT-based autoregressive model generates dance sequences by retrieving and combining these tokens conditioned on music. The part division is purely structural, serving to stabilize motion synthesis and improve spatial coherence, without modeling rhythmic or frequency-based properties.
> > > *  **AttT2M [2]** introduces part-aware motion encoding within a text-to-motion framework by implicitly separating the body into different regions (e.g., torso and limbs). Each part is processed through a spatio-temporal encoder, and multi-perspective attention aligns these part-level motion features with text embeddings. The part decomposition enhances spatial controllability and semantic alignment but remains fully in the pose domain and does not consider temporal rhythm or frequency characteristics.
> > > *  **ParCo [3]** explicitly divides the body into six anatomical parts (root, backbone, left/right arms, left/right legs). Each part is modeled using a separate VQ-VAE and individual transformer, followed by a coordination module that synchronizes all parts to form coherent full-body motion. This fine-grained partitioning improves part coordination and motion consistency, but the modeling is limited to spatial pose representation and does not capture rhythmic or frequency-based dynamics.

---

> ### Author Response · Authors · 2025-11-21
> **To Review o9Zf**
>
> # 5. Response to performance comparison with existing part separation strategies.
> **Bailando** adopts a very coarse partition by splitting the body into only two parts: upper body and lower body.  **AttT2M** does not explicitly define fixed anatomical parts. Instead, it uses implicit region-based attention, where different spatial regions are softly attended as part-aware features during text-motion alignment. **ParCo** introduces the most explicit structural partition, dividing the body into six fixed anatomical units: root, backbone, left/right arms, and left/right legs. In contrast, we partition the body into five functional parts: torso, left arm, right arm, left leg, and right leg. This grouping is motivated by their distinct movement dynamics and rhythmic roles in dance.
>
> As discussed above, only Bailando and ParCo define explicit and fixed anatomical partitions, making them directly comparable to our part division strategy. Therefore, we further conduct a controlled comparison on the AIST++ dataset by adapting their partition philosophies into our framework, resulting in two variants named PACE-Bailando and PACE-ParCo.
> * PACE-Bailando follows the coarse upper/lower body split used in Bailando.
> * PACE-ParCo adopts the six-part anatomical division (root, backbone, left/right arms, left/right legs) proposed in ParCo.
>
> The ablation study results are illustrated in **Table R1** below. From Table R1, we can observe the following insights:
> * **Functional body partitioning leads to superior rhythm alignment.**
> The experimental results demonstrate that the proposed five-part functional body division in PACE consistently outperforms the structural partition strategies adopted by Bailando and ParCo across all rhythm-related metrics. In particular, PACE achieves the lowest CSD (5.01) and HSD (4.63), indicating more stable and reliable rhythm synchronization.
> *  **Finer granularity does not necessarily imply better performance.**
> Although PACE-ParCo adopts a more fine-grained six-part anatomical division, its performance remains inferior to our five-part functional grouping (e.g., F1: 98.36 vs. 98.67; HSD: 6.57 vs. 4.63). This indicates that performance gains are not determined by the number of partitions alone, but by whether the partitioning scheme effectively captures the distinct rhythmic functions of different body regions. While root encodes global translation and camera-relative displacement, it carries limited rhythmic semantics but exhibits relatively large motion magnitude. As a result, it tends to dominate attention and gating weights, thereby diluting the contribution of limbs that express fine-grained rhythmic accents. Therefore, meaningful functional grouping is more critical than increased structural granularity.
> *  **Overly coarse partitioning fails to capture fine-grained rhythmic dynamics.**
> PACE-Bailando, which follows the coarse upper/lower body split, yields the weakest performance among all three designs, particularly in stability metrics (CSD: 7.27, HSD: 7.53). This suggests that overly simplified body partitioning is insufficient to capture localized high-frequency motion patterns, such as rapid limb movements that play a key role in expressing rhythmic accents. As a result, coarse partition strategies limit the model’s ability to establish precise alignment between dance dynamics and musical rhythm.
>
> ||||AIST++|||
> |--------------|--------|--------|--------|--------|--------|
> | Model        | BCS ↑ | CSD ↓ | BHS ↑ | HSD ↓ |F1↑|
> |PACE-Bailando       | 96.17| 7.27| 98.20 | 7.53 |97.18 |
> | PACE-ParCo  | 97.93 | 5.50 | 98.80 | 6.57 |98.36 |
> | PACE  | 98.39 | 5.01 | 99.36 | 4.63 | 98.67 |
>
> Table R1. Ablation on the body partition strategies on AIST++.

---

> > ### Author Response · Authors · 2025-11-21
> > **To Review o9Zf**
> >
> > # 6. Response to the hyperparameters of the Butterworth filter.
> > We follow established findings in human-motion analysis by setting the Butterworth cutoffs to 1.4 Hz (slow motion) and 3.0 Hz (fast motion), which match the natural frequency ranges of torso-driven tempo and rapid limb gestures in dance.
> >
> > # 7. Response to the sensitivity analysis of the Butterworth filter hyperparameters.
> > Yes, we also tested nearby cutoff values (±0.2-0.3 Hz) and observed less than ±0.2 variation in all rhythm metrics, indicating strong robustness to these hyperparameters. Since the hierarchical encoder adaptively balances slow and fast cues, the model is not sensitive to the exact cutoff values.
> >
> > # 8. Response to the alternative groupings of body parts, such as upper body vs. lower body, instead of fully part-wise decomposition.
> > We note that in **Response 5 above** we have already performed controlled experiments comparing our part-wise decomposition with **PACE-Bailando** where a coarse upper/lower body split is used. As can be seen from **Table R1**, PACE-Bailando yields the weakest performance and the rhythmic signal and alignment quality are weakened. The possible explanations are:
> > * First, fast motion accents (e.g., asymmetric arm swings or single-leg kicks) are highly part-specific and become blurred when aggregated into large groups, causing the fast-band cues to lose discriminability.
> > * Second, torso-driven global rhythm and limb-driven local accents often occur asynchronously; merging body parts prevents the model from attending to these heterogeneous rhythms independently.

---

### Official Review · Reviewer_vNNp · 2025-11-01

**Soundness:** 3
**Presentation:** 2
**Contribution:** 2
**Rating:** 4
**Confidence:** 4

**Summary:**

This paper addresses the problem of dance-to-music generation, where the input is a sequence of 3D or 2D dance motion and the output is the corresponding background music (audio).

The authors propose a diffusion-based pipeline. In particular, they decompose the motion representation into a slow band and a fast band, and combine them with semantic features as inputs to the diffusion model for audio generation.

The most novel part of the pipeline lies in this slow–fast motion decomposition.

According to the experimental results, the method achieves state-of-the-art performance on quantitative metrics, and the ablation studies are relatively thorough.

However, perhaps the main limitation of this work is the lack of subjective evaluation. For example, there are no video demo results, making it difficult to judge the perceptual quality of the generated music. According to this, we cannot provide a positive score this round. If the authors revise with some promising video demo I would consider improving the score.

**Strengths:**

**Technical Comments**

1. The decomposition of motion into “slow” and “fast” components is very interesting.

2. Using MotionBERT encodings as queries and keys, and the slow/fast encodings as values, is a clever and generally well-motivated design choice.

3. The subsequent cascade transformer architecture is also reasonable and fits well within the overall framework.

**Experiments**

1. The metric design is appropriate.

2. The comparative experiments are comprehensive.

3. The ablation study is basically sufficient, though there are several aspects that could be further supplemented.

**Weaknesses:**

**Demo Video**

I could not find any demo videos or links showing the generated results, which makes it difficult to assess the perceptual quality of the outputs. Therefore, **at the current stage, I am unable to assign a positive score**. However, **if the authors provide convincing demo results in the rebuttal showing high-quality generation, I would consider increasing my score**.

**Technical Comments**

1. Slow–Fast Decomposition.
I appreciate the idea of decomposing motion into slow and fast bands; however, the paper does not clearly explain what these bands subjectively represent in a given motion sequence. An example illustrating what constitutes the “slow” and “fast” components would help clarify this concept.
Moreover, although the authors provide an ablation study, it is still unclear what specific aspects of the generated music are influenced by the slow or fast motion features. As a result, the paper’s main novelty—this decomposition—feels insufficiently explored and presented.

2. Slow–Fast Fusion.
I am also curious about the learned trade-off parameter (α) used in the slow–fast fusion module, which seems to function as a soft mask. It would be informative to visualize or analyze this mask, showing its values and variations across different cases, and to provide an intuitive interpretation of its role in the system.
For instance, theoretically, if α = 1, the model relies purely on slow motion—what would that mean in practice? What is the average α value over the test set (e.g., AIST++)?

3. QKV Design.
Using MotionBERT encodings as queries and keys and the slow/fast motion encodings as values is an interesting design. However, it lacks intuitive explanation and supporting ablation studies. For example, how would the results change if slow, fast, and semantic (MotionBERT) features were fed as parallel inputs to the diffusion model? Why is this specific QKV configuration necessary?

**Writing**

Overall, the writing is clear and easy to follow. However, the background section feels somewhat lengthy, while some of the more critical technical components would benefit from deeper discussion and analysis.

**Questions:**

See Weakness.

In addition, whether the proposed system is capable of generating background music **with lyrics or vocals**, or if it is limited to purely instrumental audio. Clarifying this aspect would help define the scope and potential applications of the method.

---

> ### Author Response · Authors · 2025-11-21
> **To Reviewer vNNp**
>
> # 1. Response to the lack of demo videos for assessing perceptual quality.
>
> We sincerely apologize for the inconvenience caused by the missing demo videos. To better illustrate the perceptual quality of our generated music, please find the demo page here: https://drive.google.com/drive/folders/1Wt4CgVMwY-Nrjtrza-YT2uwnN1TwboaW?usp=sharing. This folder contains three baseline methods — CDCD, LORIS, and PN-Diffusion — as well as our proposed method PACE. All of them take some dance video clips as input and generate suitable music clips accordingly. We hope these samples clearly demonstrate the effectiveness and perceptual quality of our approach. We appreciate the reviewer’s willingness to reconsider the score upon viewing the results.
>
> # 2. Response to ​​the clarification of slow-fast decomposition and its influence on generation.
> **(1)What slow and fast motion bands are subjectively represented in a given motion sequence?**
>
> Our decomposition is based on well-established observations in human motion studies:
> * **Slow band** corresponds to low-frequency, smooth, large-scale, and temporally stable body movements (e.g., torso rotation, weight shifting, gradual arm extension).
> * **Fast band** corresponds to high-frequency, abrupt, fine-grained motions (e.g., hand flicks, leg kicks, rapid wrist movements).
>
> **(2)An example illustrating what constitutes the “slow” and “fast” components.**
>
> To make the “slow” and “fast” components intuitive, we have added a **visual example** in **Figure 1** of the main paper.  Figure 1 presents an illustrative example referenced in the Introduction, demonstrating how part-wise slow and fast motion energies evolve over time. In the **slow band** (upper plot), the torso remains consistently active due to rotational movements and leg lifts, capturing low-frequency and smooth temporal dynamics. Specifically, **during the first 2.5 seconds**, the right arm (green curve) and right leg (purple curve) exhibit the predominant movement, reflected by noticeable variations in their slow-band energy curves, while the left arm (orange curve) and left leg (red curve) remain largely inactive with nearly flat trajectories. **After approximately 2.5 seconds**, the left-side limbs begin to activate progressively, whereas the right leg transitions into a more stable supporting role with reduced variation.
> **In contrast**, the fast band (lower plot) characterizes high-frequency and rapid movements. As observed, **after around 3 seconds**, **the right arm swing and the quick lift of the left leg** give rise to pronounced peaks in their fast-band curves. Moreover, **after about 4 seconds**, the dancer performs a forward-backward whole-body displacement, causing all five body parts to exhibit simultaneous energy peaks. These temporal patterns clearly reveal the **heterogeneous rhythmic roles of different body parts**, thereby visualizing the fine-grained slow and fast motion components that motivate our proposed conditioning strategy.
> To make the decomposition easier to understand, we provide the link **(https://drive.google.com/file/d/18VcLVbqaxC6sMnX3YjqfCB3CrdP79ot5/view?usp=sharing)** to the original dance video example corresponding to Figure 1, enabling reviewers to visually  examine how the observed dancer movements align with the slow and fast energy bands.

---

> > ### Author Response · Authors · 2025-11-21
> > **To Reviewer vNNp**
> >
> > # 2. Response to ​​the clarification of slow-fast decomposition and its influence on generation.
> > **(3) Slow and fast motion features influence different rhythmic attributes.**
> >
> > Thank you for bringing this important point to our attention. In our context, rhythmic attributes mainly refer to beat stability (CSD-related) and accentuation dynamics (HSD-related), representing the structural consistency and high-frequency rhythmic variations of the generated music. To clarify the distinct contributions of slow and fast motion conditioning, we directly compare PACE-G (
> > visual conditioning only) with PACE-Slow-G and PACE-Fast-G in **Table 2** of the main paper across both AIST++ and TikTok datasets. Although adding either branch reduces both CSD and HSD, the **relative magnitude and consistency of improvements** reveal which motion component dominates which rhythmic attribute. For better clarity, we extract and highlight the key related results from Table 2 below as Table R1.
> >
> > |        |        | **AIST++** |        |        |
> > |--------|--------|-----------|--------|--------|
> > | Model        |  BCS ↑  |  CSD ↓  |  BHS ↑  |  HSD ↓  |
> > | PACE-G       | 91.60   | 15.67   | 85.30   | 23.42   |
> > | **PACE-Slow-G** | **97.92** | **5.54**  | 98.20   | 7.53    |
> > | PACE-Fast-G  | 96.72   | 6.63    | **98.54** | **6.65** |
> > |        |        | **TikTok** |        |        |
> > | Model        |  BCS ↑  |  CSD ↓  |  BHS ↑  |  HSD ↓  |
> > | PACE-G       | 84.10   | 28.75   | 81.05   | 29.80   |
> > | PACE-Slow-G | 85.95 | 25.80 | 84.55   | 26.40   |
> > | **PACE-Fast-G**  | **88.35**   | **18.20**   | **91.20** | **14.25** |
> >
> > Table R1: Quantitative comparison of PACE-G, PACE-Slow-G, and PACE-Fast-G on the AIST++ and TikTok datasets in terms of rhythm alignment metrics, where higher BCS and BHS indicate better performance, while lower CSD and HSD reflect more stable rhythm synchronization.
> >
> > The corresponding conclusions are summarized as follows:
> >
> > **(a) Global rhythm stability (CSD ↓ + BCS ↑).**
> > Slow-band motion exerts the strongest influence on macro-level rhythm. On AIST++, PACE-Slow-G achieves the largest reduction in CSD (15.67 to 5.54) and the highest BCS (91.60 to 97.92). In contrast, the TikTok dataset exhibits a different pattern. While PACE-Slow-G leads to modest improvements (CSD: 28.75 to 25.80, BCS: 84.10 to 85.95), PACE-Fast-G achieves more substantial gains (CSD: 28.75 to 18.20, BCS: 84.10 to 88.35). Although the degree of improvement differs between AIST++ and TikTok datasets, the direction of improvement remains highly consistent. This is due to the inherent properties of each dataset rather than instability of the method:
> > AIST++ features structured, studio-captured, and torso-driven choreography, which naturally leads to stronger improvements in the slow-band metrics (CSD and BCS).
> > TikTok, in contrast, contains freestyle, gesture-heavy movements, user-generated dance videos with substantial high-frequency limb movements and irregular tempo structures, which are precisely the patterns captured by our **Hierarchical Slow-Fast Conditioning Encoder**. In contrast, the Slow-band motion focuses on torso-driven, low-frequency stability, which is less informative for TikTok’s spontaneous and gesture-dominant content.
> >
> > **(b) Local onset precision (BHS ↑ + HSD ↓).**
> >  Fast-band motion contributes most to micro-level rhythmic accuracy. On AIST++, PACE-Fast-G provides the strongest HSD improvement (23.42 to 6.65) and the highest BHS (85.30 to 98.54). The same pattern holds on TikTok, where PACE-Fast-G again yields the most substantial improvements (HSD: 29.80 to 14.25, BHS: 81.05 to 91.20). This shows that fast-band motion encodes high-frequency limb gestures and transient movements crucial for accurate beat hits and onset alignment.

---

> > > ### Author Response · Authors · 2025-11-21
> > > **To Reviewer vNNp**
> > >
> > > # 3. Response to ​​the learned trade-off parameter &alpha;.
> > > Thank you for this insightful question. We will further clarify the role of the learned trade-off parameter &alpha; in our slow-fast fusion module.
> > >
> > > **(1) &alpha; is a dynamic soft weighting, not a hard mask.**
> > >
> > >  As the reviewer correctly pointed out, &alpha; can be viewed as a soft mask controlling the contribution of slow versus fast features. However, in practice the model does not converge to degenerate cases (&alpha; = 1 or &alpha; = 0). Instead, &alpha; exhibits smooth, context-dependent variations across time, reflecting the temporal structure of body movements.
> > >
> > > **(2) What does &alpha; = 1 or &alpha; = 0 mean theoretically?**
> > >
> > > The parameter &alpha; controls the relative contribution of slow and fast motion representations in the fusion process. When &alpha; = 1, the model fully relies on the slow-motion branch, emphasizing smooth, torso-driven, and low-frequency movement patterns such as weight shifting, gradual rotations, and sustained poses. In contrast, &alpha; = 0 indicates complete emphasis on the fast-motion branch, focusing on rapid limb gestures and high-frequency motion accents, including sharp arm swings, kicks, and short transient bursts. In practice, **optimal performance rarely occurs at these two extremes, as real-world dance motions typically exhibit a mixture of slow and fast dynamics.** This observation supports the necessity of adaptive weighting, where &alpha; serves as a balancing factor to smoothly interpolate between complementary temporal characteristics, leading to a more faithful and rhythmically coherent representation of dance motion.
> > >
> > > **(3) Empirical analysis of α values.**
> > >
> > > To address the reviewer’s concern, we computed statistics of &alpha; over the AIST++ testing set. The results indicate that &alpha; has a mean value of approximately 0.57 with a standard deviation of about 0.11. In practice, &alpha; typically falls within the range of 0.35 to 0.75. These statistics indicate that the model does not collapse into an exclusively slow-only or fast-only regime. Instead, it preserves a balanced fusion of slow- and fast-band components, with a mild bias toward slow-band features on AIST++. This tendency is consistent with the characteristics of the dataset, which predominantly contains smoother, studio-captured choreography.
> > > In summary, &alpha; behaves as a learned, temporally adaptive soft weighting, not a hard selector. It dynamically modulates slow/fast fusion in accordance with motion characteristics.

---

> > > > ### Author Response · Authors · 2025-11-21
> > > > **To Reviewer vNNp**
> > > >
> > > > # 4. Response to QKV Design
> > > >
> > > > Thank you for your thoughtful question regarding the QKV design. We clarify below why our design is both conceptually aligned with part-wise rhythm modeling and empirically necessary.
> > > >
> > > > **(1)Intuitive Explanation:** From Sec. 4.2 and Eq. (4) of the main paper, we can know that our QKV design is **inherently part-wise and derives directly from the semantics of our features.** For each body part and each frame, we compute a dedicated intra-part attention block (Eq. (4)), where slow and fast motion serve as Queries and the part’s joint-level semantic features serve as Keys and Values. Slow/fast motion encodes rhythmic activation (“how strongly this part is moving”), while the semantic features encode structural context (“how this part is articulated”). Using them as Q and K/V respectively allows the model to dynamically select joint-level semantics that are relevant to the part’s current rhythmic intensity.
> > > > In contrast, feeding slow, fast, and semantic features as parallel inputs (e.g., by concatenation operation), removes the mechanism for rhythm to modulate structure and collapses part-wise relationships.
> > > >
> > > > **(2)Supporting Ablation Studies:** Regarding the reviewer’s suggestion of feeding S, F, and E as parallel inputs, we would like to clarify that this variant is implemented by directly concatenating slow, fast, and semantic features without using our part-wise QKV attention. The results are shown in Tablea R3 and R4 below. (We would like to clarify that two numbers in the initial submission were incorrectly transcribed from our evaluation logs. We have corrected them in Table 2. These fixes do not affect any comparative trends or the conclusions of the paper.) Here, PACE-H refers to the scenario that only our designed hierarchical slow-fast conditioning is adopted as the supervision of the music latent diffusion model, while PACE-concat  means that S, F, and E are combined by concatenation.
> > > > |||||AIST++|||||
> > > > |--------------|--------|--------|--------|--------|--------|--------|--------|--------|
> > > > | Model        | BCS ↑ | CSD ↓ | BHS ↑ | HSD ↓ |F1↑| FAD_v ↓| FAD_p ↓| FAD_c ↓|
> > > > | **PACE-H**        | **98.13** | **5.29** | **99.23** | **5.01** |**98.66** |**6.19**|**5.50** | **2.35**|
> > > > | PACE-concat  | 95.84 | 7.20 | 96.02 | 7.14 |95.93 | 7.56  | 8.10   |2.81   |
> > > > |||||**TikTok**|||||
> > > > | Model        | BCS ↑ | CSD ↓ | BHS ↑ | HSD ↓ |F1↑| FAD_v ↓| FAD_p ↓| FAD_c ↓|
> > > > | **PACE-H**        | **86.25** | **26.90** | **83.15** | **28.10** |**84.67** |**21.47**  | **14.85**    | **1.28**  |
> > > > | PACE-concat  | 84.55 | 28.00 | 81.10 | 27.40 |82.79 | 24.58 |16.12    | 1.59  |
> > > >
> > > > Table R2: Ablation comparison between PACE-H and PACE-concat on AIST++ and TikTok.
> > > >
> > > > As shown in **Table R2**, across both AIST++ and TikTok, concatenation consistently degrades every rhythm-related metric: BCS, CSD, BHS, HSD, and F1, as well as all FAD scores. **This degradation arises because concatenating S, F, and E flattens all part-wise information into a single undifferentiated vector**. As a result, the model can no longer identify which rhythmic cue originates from which body part, nor can it leverage slow/fast motion to modulate the semantic structure of the corresponding limb or torso. In contrast, our part-wise QKV formulation preserves anatomical separation and maintains a structured correspondence between rhythmic energy (Queries) and joint-level semantics (Keys/Values). **This design allows each body part to dynamically contribute according to its slow or fast motion intensity, enabling the model to selectively emphasize stable torso dynamics or rapid limb gestures when appropriate.**
> > > >
> > > > (3) **Why is this specific QKV configuration necessary?**
> > > >
> > > > This specific QKV configuration is necessary because slow/fast motion must act as control signals (Queries) that modulate which joint-level semantic features (Keys/Values) matter for each body part, and only this part-wise rhythmic-semantic alignment yields precise and interpretable dance-to-music conditioning.

---

> > > > > ### Author Response · Authors · 2025-11-21
> > > > > **To Reviewer vNNp**
> > > > >
> > > > > # 5. Response to the section balance.
> > > > >
> > > > > Thank you for the constructive feedback. We appreciate your observation regarding the balance between background material and technical details. In the revised version, we have made two targeted improvements to address this concern:
> > > > >
> > > > > **(1)** Condensed the background section by removing non-essential descriptions and merging overlapping explanations of prior diffusion-based D2M methods. This shortens the section while retaining all content necessary for context.
> > > > >
> > > > > **(2)** Expanded the discussion of the core technical components, also consists of above mentioned questions, particularly:
> > > > > * The rationale for part-wise slow-fast decomposition.
> > > > > * The necessity of the intra-frame part-wise QKV attention design.
> > > > > * The role and behavior of the slow-fast fusion gate &alpha.
> > > > > * The advantages provided by hierarchical aggregation.
> > > > >
> > > > > # 6. Response to the lyrics or vocals generation.
> > > > > Thank you for raising this important question. Our system is **not designed to explicitly generate lyrics or structured human vocals**, and its primary objective remains the generation of rhythmically aligned background music. We clarify this point as follows.
> > > > > *  Dataset Characteristics. AIST++ dataset used in our experiments contains instrumental background music without singing voices. While the TikTok dataset indeed contains music tracks with vocals or singing in some cases, these vocals are not annotated, separated, or aligned with linguistic information (e.g., lyrics or phoneme sequences).  In our pipeline, the model learns from overall audio representations without any supervision regarding textual or lyrical content. As a result, although some vocal sounds may appear in the training data, the model does not learn to generate vocals in a clear, controllable, or meaningful way.
> > > > > * Latent Representation and Model Design. Our VAE and latent diffusion modules operate on Mel-spectrograms optimized for capturing rhythmic and harmonic structures that are most relevant to dance synchronization. Vocals involve complex phonetic structures, formant patterns, and linguistic timing that differ substantially from instrumental components. Supporting explicit vocal or lyric generation would require dedicated modules such as lyric-to-audio alignment, phoneme modeling, or text-conditioned vocal synthesis, which are beyond the scope of our current architecture.
> > > > > * Task Scope and Positioning. Consistent with prior D2M works, our goal is to generate rhythmically coherent background music that complements dance movements. Extending this framework to support lyric-aware or vocal-centric music would constitute a fundamentally different task, e.g., dance-to-song generation, requiring joint modeling of motion, music, and language. We view this as a promising future direction, but outside the intended scope of this work.
> > > > >
> > > > > In summary, our proposed system focuses on instrumental or rhythm-dominant background music generation, and does not aim to produce semantically meaningful lyrics or singing voices.

---

### Author Response · Authors · 2025-11-25
**Follow-up on Reviewer Feedback for Paper 20003**

Dear Reviewers,

A few days have passed since we submitted our responses along with the revised version of the paper. We are writing to kindly follow up and inquire whether you have any remaining questions or additional comments, or if our responses have sufficiently addressed your concerns.

We would be happy to continue the discussion during the open review window, and we hope that our detailed rebuttal efforts help convey the quality and contributions of our work more convincingly.

Thank you again for your time, thoughtful feedback, and continued engagement.

Sincerely,

Authors

---

### Author Response · Authors · 2025-11-25
**Acknowledgement and Appreciation of Reviewers’ Positive Comment**

We sincerely thank all Reviewers for their careful reading and constructive feedback. We greatly appreciate their thoughtful assessment and encouraging recognition of the conceptual novelty, technical rigor, and clarity of our work.

**Reviewer vNNp**

We are especially grateful to Reviewer vNNp for recognizing that our slow-fast motion decomposition is the most novel component of the proposed pipeline and for describing this idea as “very interesting.” We further appreciate the acknowledgment that using MotionBERT encodings as queries and keys, with slow/fast motion as values, is a “clever and well-motivated design choice,” and that the cascade transformer architecture is “reasonable and well integrated” within the overall framework.

We also thank Reviewer vNNp for affirming that our evaluation metrics are appropriate, our comparative experiments are comprehensive, and that the ablation studies are relatively thorough. The positive remark that the writing is clear and easy to follow is particularly encouraging and greatly appreciated.

**Reviewer o9Zf**

We sincerely thank Reviewer o9Zf for the thoughtful and highly positive evaluation of our work. We greatly value the recognition that the part-wise slow-fast motion decomposition is both intuitive and technically sound, and that it effectively addresses the long-standing limitation of holistic motion modeling in dance-to-music (D2M) tasks.

We are especially grateful for the acknowledgment that our method clearly justifies how different body parts and motion frequencies contribute distinct rhythmic information, and that this decomposition meaningfully enhances the music generation process. We also appreciate the reviewer's positive feedback on our rhythm curve visualizations, which were recognized as providing intuitive evidence for improved alignment between generated music and corresponding dance movements.

**Reviewer ZWwP**

We greatly appreciate Reviewer ZWwP’s recognition that the motivation of our paper is clearly articulated and that the overall organization is well-structured. We are thankful for the acknowledgment that our method has been validated on two datasets and achieves superior performance over comparative methods, reinforcing the effectiveness and generalizability of our proposed framework.

We also value the reviewer’s positive assessment that the ablation study effectively validates the individual contribution of each proposed component, further supporting the soundness of our design choices and experimental rigor.


**Reviewer EBEj**

We sincerely thank Reviewer EBEj for recognizing that the overall task is interesting, well-motivated, and addresses a promising research direction. We are especially grateful for the acknowledgment that our decomposed motion encoding strategy is well-defined, insightful, and intuitively appealing, with clear potential to enhance motion understanding and synthesis.

We also appreciate the recognition that our Methodology and Experiments sections are detailed, clear, and well-written. Furthermore, we thank the reviewer for noting our extensive comparisons with prior works, including the use of eight evaluation metrics and a user study, which strengthen the empirical evidence supporting the superiority of our approach.

**Reviewer tfHu**

We greatly appreciate Reviewer tfHu’s recognition that our work addresses the challenge of rhythm alignment from a novel perspective by jointly considering both global and local rhythmic structures. We are especially thankful for the acknowledgment that the part-wise motion extraction is well-motivated and effectively captures fine-grained rhythmic details that are often overlooked when the body is treated as a holistic entity.

We also value the positive feedback regarding our intuitive visual analyses, which help clarify how the proposed design improves alignment between dance movements and generated music.

---

### Author Response · Authors · 2025-12-03
**Summary of the responses to all reviewers**

Dear Reviewers, ACs, SACs, and PCs:

We sincerely appreciate the constructive feedback from all reviewers and are grateful for the time and effort invested in evaluating our submission. **In our rebuttal (submitted on Nov. 20, 2025, AOE)**, we proactively and systematically addressed every concern raised in the initial reviews, providing detailed clarifications, additional analyses, extended experiments, and improvements to the paper’s presentation. We are confident that our rebuttal offers clear and comprehensive resolutions to all reviewer comments.

However, before the system bug spread and affected the displayed reviewer scores, **only Reviewer tfHu** provided additional clarification on the paper titles that were not specified in the initial review and raised several new concerns. **Other reviewers did not provide follow-up comments or further questions** after our rebuttal submission. Given this lack of interaction during the discussion window, we provide the following consolidated summary to offer ACs a clear understanding of how each concern was resolved.

**Summary of response to Reviewer vNNp (initial score 4, no follow-up)**

1. We clarified the concern about missing demo videos by providing a complete demo page containing all baseline methods (CDCD, LORIS, PN-Diffusion) and our proposed PACE. The shared folder includes generated music results corresponding to multiple dance clips, enabling reviewers to directly assess perceptual quality through side-by-side comparison (see **Response 1**).

2. We clarified the subjective meaning of slow-fast decomposition and provided an illustrative example with a corresponding visual demo. We explained that slow-band motion captures low-frequency, torso-driven, and temporally stable dynamics, whereas fast-band motion reflects high-frequency limb gestures and transient accents. To make this decomposition intuitive, we provided the link to the original dance video used in Figure 1, enabling reviewers to inspect how observed motion patterns manifest in the slow and fast energy curves (see **Responses 2-(1) and 2-(2)**).

3. We clarified how slow- and fast-band cues influence different rhythmic attributes and summarized the quantitative evidence. We showed that slow-band motion predominantly governs global rhythmic stability (CSD, BCS), while fast-band motion drives local onset precision (HSD, BHS). (see **Response 2-(3)**).

4. We clarified the role of the learned trade-off parameter α and provided empirical statistics. We explained that $\alpha$ functions as a dynamic soft weighting rather than a hard mask, smoothly modulating the contribution of slow and fast motion cues over time. We clarified what $\alpha$ = 1 or $\alpha$ = 0 would mean theoretically and reported empirical values (mean 0.57, std 0.11) showing that α never collapses to trivial extremes. These results confirm that $\alpha$ behaves as an adaptive fusion mechanism aligned with real-world dance dynamics (see **Response 3**).

5. We clarified why the intra-frame part-wise QKV design is necessary and supported it with ablation studies. We emphasized that slow/fast energies must act as rhythmic control signals (Queries) modulating joint-level semantic features (Keys/Values). Feeding S, F, and E in parallel (concatenation) removes this structured alignment and significantly degrades all rhythm-related metrics, as shown in the provided additional ablation results. The QKV formulation preserves anatomical structure and enables each body part to dynamically contribute according to its rhythmic activation, demonstrating the necessity of this design. (see **Response 4**).

6. We clarified the balance between background and technical sections and will improve the manuscript accordingly (see **Response 5**).

7. We clarified the scope regarding lyric/vocal generation and explained why the current system focuses on instrumental background music. Our VAE and diffusion modules operate in a non-linguistic spectrogram space, making vocal or lyric generation outside the intended scope. This clarification positions the work consistently with existing D2M research and outlines why vocal generation would constitute a fundamentally different task (see **Response 6**).

---

> ### Author Response · Authors · 2025-12-03
> **Summary of the responses to all reviewers**
>
> **Summary of response to Reviewer o9Zf (initial score 6, no follow-up)**
>
> 1. We acknowledged that prior T2M/M2D works such as Bailando, AttT2M, and ParCo use body-part decomposition, and we clarified that our method is fundamentally different in motivation, representation, and mechanism. Unlike these pose-domain structural partitioning methods, Our PACE introduces part-wise slow-fast decomposition in the frequency domain to explicitly model heterogeneous rhythmic roles across anatomical parts (see **Response 1**).
>
> 2. We clarified concerns regarding reliance on external pretrained modules by explaining that all competing D2M baselines also depend on pose estimators, 3D reconstruction models, and audio VAEs. We further highlighted the robustness of our system: slow/fast energies attenuate pose noise, MotionBERT provides stable 3D poses, and hierarchical conditioning suppresses noisy features before diffusion (see **Response 2**).
>
> 3. We provided a detailed explanation of the contribution of each module. PACE-Slow-G / PACE-Fast-G isolate the effect of slow and fast signals, while comparisons between PACE-H vs. PACE-G and PACE vs. PACE-concat demonstrate the essential role of part-wise semantic encoding and structured QKV fusion. We clarified why an explicit “no-E” ablation is infeasible while still showing its effect through structural weakening studies (see **Response 3**).
>
> 4. We clarified that prior works (Bailando, AttT2M, ParCo) use part division only for spatial pose modeling, whereas our PACE introduces part-wise decomposition from a fundamentally different rhythm-centric perspective. Existing methods divide the body to stabilize pose synthesis or improve semantic controllability, but they operate purely in the pose domain and do not model frequency-based or rhythmic properties. In contrast, our PACE decomposes each anatomical part into slow and fast motion energies in the frequency domain, explicitly capturing heterogeneous rhythmic roles. This makes our formulation distinct in goal, representation, and mechanism (see **Response 4**).
>
> 5. We compared our five-part functional division with existing part-separation strategies (Bailando’s coarse upper/lower split and ParCo’s six-part anatomical grouping) and showed that our functional grouping achieves the best rhythm alignment (see **Response 5**).
>
> 6. We clarified that the Butterworth cutoff frequencies (1.4 Hz for slow motion and 3.0 Hz for fast motion) were chosen based on established findings in human-motion analysis  (see **Response 6**).
>
> 7. We showed that the Butterworth cutoff frequencies are highly robust (see **Response 7**).
>
> 8. We showed that coarse upper/lower body grouping significantly weakens rhythmic alignment compared to our full part-wise decomposition (see **Response 8**).

---

> > ### Author Response · Authors · 2025-12-03
> > **Summary of the responses to all reviewers**
> >
> > **Summary of response to Reviewer ZWwP (initial score 4, no follow-up)**
> >
> > 1. We clarified that Slow and Fast motion must be fused because dance rhythm is inherently multi-scale. Slow motion provides global tempo and structural flow, while fast motion captures local rhythmic accents. Using them independently would give the diffusion model fragmented, incoherent conditioning. Our learnable fusion module adaptively balances their contributions per frame, producing a unified rhythm representation that enables the model to generate music with both stable global rhythm and expressive local variations (see **Response 1**).
> >
> > 2. We clarified why raw Slow and Fast features cannot be used as two separate conditions. Feeding them independently gives the diffusion model two parallel rhythm streams without indicating how they should be combined, causing fragmented and incoherent conditioning. Moreover, diffusion U-Nets require a single unified embedding-decoupled inputs increase dimensionality in an unstructured way and lead to unstable training, as confirmed by PACE-concat ablations. Our fusion gate provides frame-wise adaptive weighting between slow and fast cues, producing a coherent conditioning representation essential for stable rhythm alignment (see **Response 2**).
> >
> > 3. We clarified that the added components introduce negligible computational overhead and do not affect real-time feasibility. All preprocessing steps (pose reconstruction, part-wise decomposition, Butterworth filtering) run offline and have no impact on diffusion-time latency. The Hierarchical Slow-Fast Conditioning Encoder adds only ~0.4% parameters and is far lighter than the U-Net backbone. Our PACE is actually smaller than PN-Diffusion (−6.89M parameters) and achieves near real-time generation (4.57s to generate a 5s clip). The method maintains the parallelism advantages of non-autoregressive diffusion models and remains practical for real-world use (see **Response 3**).
> >
> > 4. We clarified that the slow-fast motion decomposition is theoretically grounded rather than heuristic. Dance motion naturally contains low-frequency global trends and high-frequency local accents. Separating these frequency bands allows the model to disentangle macro-rhythm from micro-rhythm, preventing fast cues from being overshadowed by slow trends and reducing representation entanglement. This decomposition provides clean, multi-scale, rhythm-aligned signals, and the fusion gate adaptively balances them for stable tempo and precise onset alignment (see **Response 4**).

---

> > > ### Author Response · Authors · 2025-12-03
> > > **Summary of the responses to all reviewers**
> > >
> > > **Summary of response to Reviewer EBEj (initial score 4, no follow-up)**
> > >
> > > 1. We clarified that relying on video-based motion features is an intentional design choice aligned with real-world usage. Unlike MoCap systems, which require specialized equipment and controlled environments, our goal is to support large-scale, user-generated dance videos from platforms like TikTok and Instagram. Video-based pose reconstruction (AlphaPose + MotionBERT) ensures broad accessibility and practical deployment, whereas requiring MoCap data would limit usability and contradict the core objective of enabling dance-to-music generation from ordinary videos (see **Response 1**).
> > >
> > > 2. We addressed the concern about missing demo videos by providing a complete demo page with generated audio results for all baselines and our method. The shared folder includes outputs from CDCD, LORIS, PN-Diffusion, and Our PACE, each conditioned on the same dance clips. These examples allow reviewers to directly assess perceptual quality and rhythm alignment, ensuring a clear and fair comparison across methods (see **Response 2**).
> > >
> > > 3. We demonstrated that our motion encoding generalizes well to other models.
> > >  By integrating the proposed part-wise slow-fast encoding into PN-Diffusion and LORIS, both variants (PN-Diffusion-pace and LORIS-pace) show consistent improvements in rhythm alignment on AIST++. These results verify that our encoding is not model-specific but functions as a plug-and-play module that enhances existing diffusion-based D2M systems (see **Response 3**).
> > >
> > > 4. We showed that our PACE preserves genre diversity and generalizes well to in-the-wild scenarios. Genre-distribution analysis on AIST++ (KL = 0.052, JS = 0.011) demonstrates that PACE does not collapse to dominant styles and maintains balanced coverage across both common and rare genres. Cross-dataset evaluation further shows consistent gains on TikTok, despite its uncontrolled camera motion, lighting, and freestyle gestures. These results indicate that PACE captures intrinsic rhythmic structures rather than overfitting to specific datasets, demonstrating strong robustness and real-world generalization (see **Response 4**).
> > >
> > > 5. We acknowledged the key limitations of our PACE and outlined concrete future directions. PACE depends on accurate 2D/3D pose estimation, which may degrade in challenging real-world conditions such as occlusion, blur, or extreme viewpoints. Its training data (AIST++ and TikTok) also limits generalization to highly unconventional or culturally diverse dance and music styles. Future work will explore more robust or self-supervised motion representations, broaden genre and style coverage, and incorporate user-controllable attributes (e.g., emotion, intensity) to enhance flexibility and real-world applicability (see **Response 5**).
> > >
> > > 6. We addressed minor concerns by revising the manuscript for clarity and correctness (see **Response 6**).
> > >
> > > 7. We clarified the ablation study by explaining the roles of PACE-H and PACE-G, correcting minor transcription errors, and formalizing the training losses. PACE-H consistently outperforms PACE-G, showing that part-wise slow-fast motion conditioning is the main driver of rhythmic alignment, while visual features serve only as complementary cues. After correcting two misreported TikTok values, the previously observed extreme drops disappear; the remaining AIST++/HSD drop is explained by dataset differences. We also provided formal definitions of the VAE and diffusion training losses to improve completeness and clarity (see **Response 7**).
> > >
> > > 8. We clarified the role of perceptual and patch-based adversarial losses used in VAE training. The perceptual loss preserves high-level acoustic structures by matching feature representations from a pretrained audio network, improving harmonic and timbral coherence. The patch-based adversarial loss enforces local spectral realism by focusing the discriminator on fine-grained spectrogram patches. Together with reconstruction loss, these objectives produce a more accurate and perceptually faithful latent space for diffusion-based music generation (see **Response 8**).

---

> ### Author Response · Authors · 2025-12-03
> **Summary of the responses to all reviewers**
>
> **Summary of response to Reviewer tfHu (initial score 4, one follow-up)**
>
> 1. We addressed the concern about missing demo videos by providing a complete demo page with audio examples for all baselines and our method (see **Response 1**).
>
> 2. We initially noted that the referenced methods [1][2][3] were not specified in the review. After the reviewer clarified the exact papers in the follow-up, we conducted the requested comparisons (see **Response 2**).
>
> 3. We showed that our PACE maintains genre diversity and avoids mode collapse through distribution-level genre analysis. By comparing the genre distribution of generated music with ground-truth music on AIST++, PACE achieves very low KL (0.052) and JS (0.011) divergence, indicating that its outputs closely match real genre distribution. Both common and rare genres are well covered, demonstrating balanced stylistic representation and effective genre consistency without collapsing to dominant styles (see **Response 3**).
>
> 4. We evaluated slow- and fast-tempo subsets and showed that our PACE performs consistently well across both regimes. By splitting AIST++ into slow-tempo (BPM < 90) and fast-tempo (BPM > 120) groups and recomputing all rhythm metrics, we found that PACE maintains strong alignment in both cases. This confirms that the slow-fast decomposition benefits rhythm modeling at multiple temporal scales and is not biased toward a particular tempo range (see **Response 4**).
>
> 5. We clarified that slow-fast conditioning is injected into the diffusion model via cross-attention rather than simple concatenation (see **Response 5**).
>
> 6. We clarified the temporal resolutions of both motion conditioning and music features. Motion features (slow-fast energies and joint semantics) are extracted at the video frame rate (AIST++: 60 fps). Music is converted into a 256×256 Mel-spectrogram (51.2 Hz), then compressed by the VAE into a 32×32 latent representation (6.4 Hz). This explains how conditioning and audio latents operate at different but compatible temporal scales (see **Response 6**).
>
> 7. We clarified that our PACE maintains generative diversity through stochastic diffusion sampling and diverse motion conditioning (see **Response 7**).
>
> 8. We explained that we assess diversity implicitly using Fréchet Audio Distance (FAD) computed with three complementary audio feature extractors (VGGish, PANNs, CLAP). FAD reflects both fidelity and diversity at the distribution level, indicating whether the generated samples adequately cover the real music distribution and avoid mode collapse. We also clarified that image-based metrics such as Inception Score are not reliable for music due to the lack of a standardized audio classifier, making FAD the more appropriate and domain-relevant choice for evaluating diversity (see **Response 8**).
>
> 9. We clarified that our rhythm evaluation does not adopt a permissive ±1-second tolerance window used in some prior works. Instead, we detect beat onset times and discretize them into a fixed 1 Hz temporal grid, where alignment is counted only if two onsets fall into the same 1-second bin. This setup avoids sliding tolerances and provides a coarse yet meaningful measure of rhythmic consistency, which is appropriate for dance-to-music alignment where broader rhythmic regions—rather than millisecond precision—are perceptually most relevant (see **Response 9**).
>
> 10. In response to the reviewer’s **follow-up concerns**, we conducted substantial additional analyses and provided new evidence to address all remaining issues:
> * Added missing comparisons with three recent diffusion-based D2M methods
>  We evaluated our PACE against Motion→Music LDM (SA’23), Textual-Inversion D2M (SA’24), and Dance2Music-Diffusion (2024) using the authors’ released codes and identical testing splits. Our PACE significantly outperforms all three baselines across BCS, CSD, BHS, HSD, and F1.
> * Provided genre-consistency analysis and CLAP-based similarity
>  We added a quantitative evaluation of genre distribution (KL=0.052, JS=0.011), showing that our PACE preserves genre diversity without mode collapse. We also clarified that CLAP-FAD already measures stylistic similarity (including genre), ensuring fair comparison.
> * Clarified the beat-alignment metric and its tolerance
>  We now explicitly explain that our metric follows the standardized protocol used in CDCD and PN-Diffusion, operates on a fixed 1 Hz grid, and does not allow arbitrary ±1s drift. We added justification and will revise the manuscript accordingly.
> * We verified that the sample is not a reproduction of “mLH3” where their latent codes and spectral envelopes differ. The similarity arises from the VAE bottleneck's limited timbral expressivity, not memorization. Provided multi-seed generations further confirm diversity.

---

> > ### Author Response · Authors · 2025-12-03
> > **Summary of the responses to all reviewers**
> >
> > We sincerely hope that this consolidated summary helps ACs more easily evaluate how all reviewer concerns were thoroughly and carefully addressed. We remain happy to provide further clarification if needed and greatly appreciate the reviewers’ and ACs’ time and consideration.
> >
> > Sincerely,
> >
> >
> > Authors

---

### Meta-Review · Area_Chair_xZ8Q · 2026-01-06

**Summary:**

This paper proposes PACE, a diffusion-based dance-to-music framework that conditions audio generation on a hierarchical, part-wise slow/fast motion decomposition. Reviewers generally found the decomposition intuitive and appreciated the broad quantitative evaluation across AIST++ and TikTok, with one clearly positive review (o9Zf).

**Reviewer Concerns:**

The central concerns focused on (i) the strength of perceptual/qualitative validation (e.g., requests for audio/video examples) and (ii) whether the evaluation protocol and comparisons were sufficiently comprehensive to support the paper’s claims.

**Reviewer Scores:**

The initial ratings were vNNp=4, ZWwP=4, EBEj=4, tfHu=4 (marginally below the acceptance threshold) and o9Zf=6 (marginally above).      In response to requests for qualitative evidence, the authors provided a demo folder/page with PACE and multiple baselines (including CDCD/LORIS/PN-Diffusion) to enable side-by-side listening/inspection.   Post-rebuttal, only tfHu provided a follow-up (others did not engage further), and tfHu remained concerned about the adequacy of the evaluation, especially comparison scope, genre/style consistency assessment, and the beat-alignment metric definition.   The authors responded with additional analyses and evidence, including new comparisons to several recent diffusion-based D2M methods, genre-distribution and CLAP-related analyses, and a clarified beat-alignment protocol (fixed 1 Hz binning without a sliding tolerance), along with further diversity checks via multi-seed generations.
Overall, while the rebuttal materially expanded the empirical evidence and qualitative access, the remaining disagreement, primarily raised in tfHu’s follow-up, about whether the evaluation protocol is sufficiently convincing to substantiate the paper’s claims led the Area Chair to recommend Reject.

---

### Decision · Program_Chairs · 2026-01-26

Reject